# A robotic prebiotic chemist probes long term reactions of complexifying mixtures

Silke Asche[1], Geoffrey J. T. Cooper[1], Graham Keenan[1], Cole Mathis[1] & Leroy Cronin [1✉]

To experimentally test hypotheses about the emergence of living systems from abiotic chemistry, researchers need to be able to run intelligent, automated, and long-term experiments to explore chemical space. Here we report a robotic prebiotic chemist equipped with an automatic sensor system designed for long-term chemical experiments exploring unconstrained multicomponent reactions, which can run autonomously over long periods. The system collects mass spectrometry data from over 10 experiments, with 60 to 150 algorithmically controlled cycles per experiment, running continuously for over 4 weeks. We show that the robot can discover the production of high complexity molecules from simple precursors, as well as deal with the vast amount of data produced by a recursive and unconstrained experiment. This approach represents what we believe to be a necessary step towards the design of new types of Origin of Life experiments that allow testable hypotheses for the emergence of life from prebiotic chemistry.

[1] School of Chemistry, University of Glasgow, Joseph Black Building, University Avenue, Glasgow, UK. ✉email: lee.cronin@glasgow.ac.uk

On the early Earth, prebiotic chemistry underwent a transition to biological chemical systems during a very long period (ca. 100 Myr)[1], yet explorations in the laboratory today are traditionally limited to a few hours or, at most, days. Only 3.7% of all experiments reported to Reaxys between 1771 and 2011 were carried out for longer than 2 days[2]. However, the exploration of unconstrained[3] or complex multi-component systems requires far longer times and a large number of parallel experiments[4,5], after which progress is hindered by the analytical complexity of the products; huge numbers of samples containing unknown mixtures[6], seen by many as intractable. This is further complicated by the fact that realistic chemical reactions, vital to emulate the types of processes possible on the early Earth at the Origin of Life, did not take place in a clean single environment.

There are many candidate theories and frameworks that aim to explain how living systems can emerge from nonliving substrates[7–9], but none of these are testable over the long time periods over which life was thought to have emerged on Earth ca. 3.8 B years ago[10]. For example, it has long been hypothesized that the central carbohydrate metabolism emerged as a geochemical process without enzymes and subsequently evolved via the addition of ever more complex reaction pathways[11,12]. While component pieces of this idea have been tested, the entire hypothesis cannot be explored using current technology. The same is true of many other hypotheses regarding the origin of cellular membranes and genetic molecules[13,14]. This exposes an important gap that can be explored. Much current research focuses on prebiotic plausibility, which itself is constrained by our geochemical knowledge[15], and a vast amount is simply unknown, namely the space of chemical reactions and starting materials available, as well as the precise reaction conditions and constraints on these conditions. Previous approaches to prebiotic chemistry, which have their origin in synthetic organic chemistry, intentionally try to limit the accessible size of the chemical space in experiments[16,17]. While this is convenient, as it allows the identification of individual products using standard analytical techniques, experimental conditions need to move away from these single-flask approaches to include controlled environmental factors if we are to explore the chemical space relevant to the emergence of living systems. This can include, for example, the inclusion of mineral surfaces and variable temperature, pH, and redox conditions, some or all of which may be allowed to vary dynamically, driven by, and driving, the chemical reactions in the mixture. Some work has already shown that chemical reactions of simple 'soups' in cycles lead to the diversification and differentiation in the product space[3,5,18,19], but the number of potential reactions and time needed for all the reactions is vast[18,20–22].

Currently the field lacks an experimental design framework that would allow researchers to test competing hypotheses[23] on long timescales, and the number of candidate experiments is gigantic[24,25]. This problem is made even bigger when the vastness of search space relevant for the investigation of the emergence of life is considered—such a chemical space cannot be adequately explored using experiments that run for a day or a few hours. Here we show a 'robotic prebiotic chemist', an automated closed-loop system that runs unconstrained multicomponent chemistry experiments on mineral surfaces in cycles, with fully automated analytical measurements and a decision-making metric.

## Results and discussion
In this work, we set out to design a system that cannot only automate a vast number of experiments, but also make decisions on the fly about which routes to follow. We wanted to design an experimental platform that could be automated and have a range

of input reagents, heterogeneous reaction environments, and the ability to carry out reaction cycles recursively over a very long period of time (weeks to months). In aiming for such an experimental design, we also wanted to include a sensitive in situ assay that would allow our robotic prebiotic chemist to search chemical space autonomously[26]. Thus, our goal was to search for molecules of increasing complexity as a function of the product distribution and cycle number, and follow the changes in these distributions, all in a reproducible way[27]. In our experiment, we intended to find a rise in the complexity or information content of our complex mixture over cycles, looking for an increase in the mass of the product species by the use of the algorithm.

To help understand the experimental design constraints, we first explored the chemical space accessible to the input reagents computationally, see Fig. 1, using the Molecular Transformer[28] to

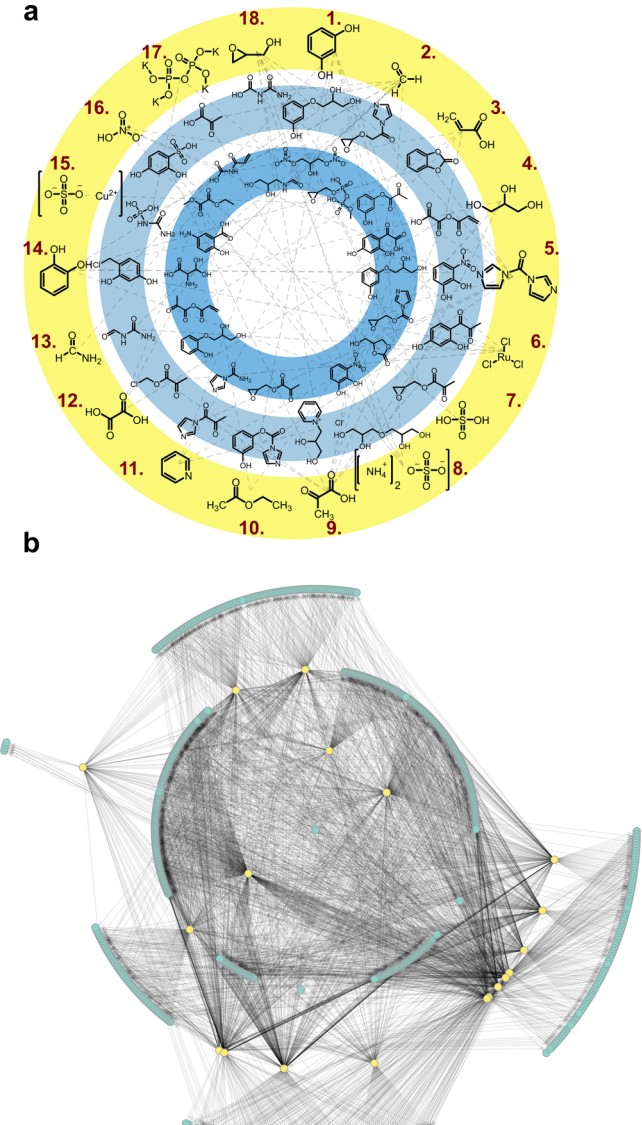

**Fig. 1 Network of reactions possible based on the input library.** On the top panel **a** is a network where compounds of the starting material library (yellow, outer circle) are reacting to example products with either two starting compounds (pale blue, middle circle) or three compounds (dark blue, circle inside) reacting at once. On the bottom panel **b** is a full network of all known possible reactions shown, again with two or three input molecules reacting together. Input compounds are represented as yellow dots, while all found reaction products are presented as blue dots.

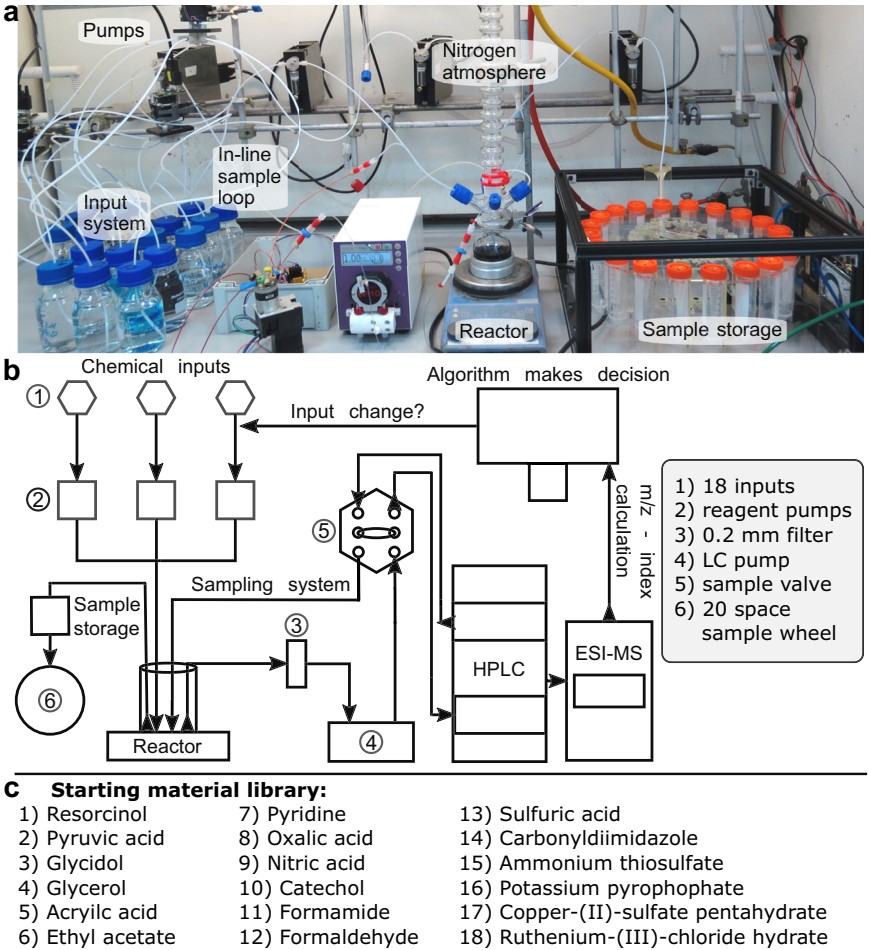

**Fig. 2 Platform overview. a** A photograph of the platform set up is shown above. The analytical system, including HPLC–MS and the computer, which controlled the platform processes, is not shown in the photograph. **b** A detailed schematic depiction is shown in the middle. **c** A list of the starting material library is shown below.

test all combinations of two and three inputs from the pool of 18 possible input reagents.

The selection of these initial 18 input molecules, which are shown in Fig. 1, was by a human and inevitably is biased. However, within the set of 18 materials, the compositions of inputs for the experiments are chosen at random or by the algorithm. It can be argued that the input decision is the most important decision for the experiment, but as the total chemical space is so vast, this preselection is necessary to contain the possible options. We tried to concentrate on the function of the building blocks themselves, rather than taking any prebiotic assumptions into account (no "common" autocatalytic cycle precursors, sugars, or amino acids are included). Further information about the selection and properties of each building block can be found in the SI section 2.

Briefly, the molecular transformer is a machine-learning model that takes input reagents as arguments and suggests possible product species, the model has a built-in mechanism to assess (or score) the quality of its prediction on a scale of 0.0–1.0, with reactions scored as 1.0 being the most supported by observed reactions. For our purposes, any combination that gave a score greater than 0.8 was saved and the candidate product was recorded. This analysis yielded 2206 possible reactions, which means they can be predicted based on the structure of the reactants and previously reported reactions from the literature. To visualize this chemical space, these reactions were represented as

a network, with reactants being connected to the products if they are part of the same reaction, see Fig. 1B[29].

This shows that when two of the three input reagents are reacted together in water, a wide range of products are possible. This simulation does not take reagent concentration or experimental conditions into account, but through the representation of possible reactions, the potential of such an experiment to create a "chemical mess" becomes clear. Using this analysis, it is clear that an innovative analytical workflow will be required to analyze such a complex product mixture.

The platform, which is shown in Fig. 2, can execute experiments with changing input compositions, handling heterogeneous mixtures of liquid and fine-ground solids, and running a continuous experiment with several hundred cycles for more than 30 days with minimal human intervention. The platform comprises five pumps (four syringe pumps and one peristaltic pump), four valves for liquid handling (three for inputs, one for analysis), 18 reagents in aqueous solution, a reactor vessel under nitrogen and fitted with a reflux condenser, an IKA computer-controllable magnetic stirrer hot plate, a sampling loop for HPLC–MS, and a purpose-built sample wheel to store up to 20 samples for offline analysis. The only part of the platform to not be automated was the manual changing of the vials in the sample wheel every 3 days. Three of the pumps and valves are used to connect the reagent bottles to the reactor. Another pump is used to deliver the product mixture from the reactor to the sample wheel. The

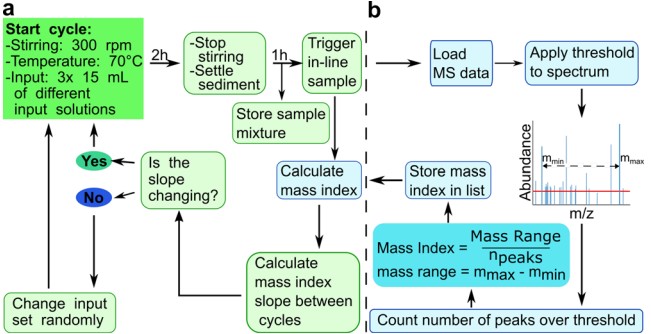

**Fig. 3 Schematic workflow of the automated system. a** The general tasks. **b** The decision-making process of the algorithm. The formula for the Mass Index was previously reported by Doran et al.[20].

nitrogen atmosphere was a deliberate choice to avoid potential oxidation of the reaction products and to be able to control the experimental atmosphere. The online analytical system is supported by the peristaltic pump, which takes liquid from the reactor, pulls it through a syringe filter, and injects it into an external HPLC valve, which is flushed with the mobile phase directly onto the column and from there to the ESI-MS. Python code is used to control the platform and analyze the MS data.

The system can change experimental conditions and inputs based on the data acquired during the experiment. Each experimental run on this platform consists of 60–150 cycles with cycle times from 3 to 12 h and started with a clean dry reactor, charged with a mixture of freshly washed minerals (quartz ($SiO_2$)), ulexite ($NaCaB_5O_6(OH)_6 \cdot 5H_2O$ and pyrite ($FeS_2$)). At the beginning of each experiment, 30 mL each of three randomly assigned input solutions were added to the reactor. Stirring and heating were set to 70 °C and 300 rpm. At the end of the cycle, heating and stirring were stopped and the system was allowed to settle for 1 h to prevent mineral particles from blocking the hardware while sampling. A sample was then taken into the online analysis sample loop, and 70% of the total product solution was removed to a vial in the sample wheel for storage and offline analysis. The remaining mineral slurry and product mixture were then replenished with fresh input solutions before stirring/heating was restarted to begin the next cycle. By recursively cycling and diluting the product mixture with the addition of further starting reagents, our experiments were designed to dilute out any product compounds that are not robust during the reaction cycle. Thus, the only compounds remaining in significant abundance after many cycles would be those that were produced over the cycle, persisted over dilution, or were bound to the mineral surfaces. The workflow for this system can be found in Fig. 3.

We performed experiments using an algorithm that chose three of the 18 building blocks randomly for each input composition. The experiments aimed to find a way to build increasing complexity in these unconstrained (by which we mean, in this context, that the system is not restricted within the compound library, to change the input composition at any given time without human intervention/constraint) multicomponent chemical mixtures and enable analysis to track the experiment on the fly, permitting a change of the experimental inputs if sufficient change in the product mixture could not be observed.

As shown in Fig. 1, the selected input reagent library is prone to create a messy chemical mixture, which is a chemical system known to be too complex to be analyzed conventionally[6,30,31], with identification of all the product species involved. We address the difficulty of analyzing each specific product in these mixtures by taking the alternative 'systems' approach of observing the behavior of the chemical mixtures and looking for global

phenomena in the recorded data, rather than concentrating on targeted analysis. The vast amount of collected data adds to this problem, as the algorithm must be able to access, analyze, and interpret the result of the previous cycle quickly in order to make any necessary adjustments for the following one. The product mixture was directly analyzed via LC–MS and the algorithm assigned the outcome of the cycle automatically, making the input decision for the next cycle as shown in Fig. 3. The MS data are automatically analyzed using the 'Mass Index' metric, which was reported previously and is explained in Fig. 3[20]. The total ion chromatogram is accessed with code that extracts each single spectrum and searches for the heaviest and the lightest peaks over a threshold of $10^6$ intensity. The peaks are subtracted from each other and divided by the number of peaks over threshold. The result of this calculation can be used as a label for each cycle, enabling automated comparison. The resulting number is stored in a list and the slope between the previous cycles is calculated. If the slope is below zero and a minimum number of cycles have been executed with the same set of inputs (usually a set of inputs was run for 10 cycles before a new decision was made), the algorithm decides to change the input composition randomly from the library of input solutions.

The Mass Index value sets the highest and lightest peak in relation with the total number of peaks. If there was one heavy peak and just a few other product species, the number would be high. The more peaks are detected and the lighter the heaviest peak, the lower the Mass Index value would be. This approach can detect a combinatorial explosion, in which the Mass Index would be very low, caused by the number of detected species. Furthermore, the index can detect the development of a dominant heavy species, as the Mass Index rises with the occurrence of heavier peaks, while the number of total species would be low (see supplementary table 3.2 for some examples).

By calculating the slope between Mass Index values of different cycles, we can determine if the experiment changed in-between cycles. Thus, the Mass Index provides a simple heuristic we can use to evaluate MS data without human intervention enabling the automated system to change the input composition based on the experimental outcomes. Further information about the decision-making process can be found in the SI, section 5.3. Experimental data of the Mass Index and the corresponding slope between the cycles in which the input composition was changed is shown in Fig. 4. There is no immediate trend observable in the overview of the presented experiments, but four of the six experiments show periods where the Mass Index progressively increases, even as the selected inputs are changed. This means that a heavier product was forming, and/or the number of overall peaks over threshold was declining, while the heaviest mass remained. In the other two experiments, there is no trend to notice and the slope is rather stable or shows just small changes. The algorithm was designed to change the experiment as soon as the Mass Index of the product mixture stopped increasing, with the goal of pushing the system back out of a steady state or equilibrium. Using this method, we expected that the data would result in a progressive increase in the Mass Index over cycles. Interestingly, the results in Fig. 4 show that this is not universally true. We can observe rising trajectories and trajectories that seem to alternately rise and fall. For example, run C (Fig. 4) is nearly static, while run A and F show significant increases in the Mass Index. One explanation for the static Mass Index values in run C could be an unfavorable chemical composition that is not prone to react, together with the fact that the change and the random input decision did not lead to any complex or changing product mixture. Run A has the highest observed Mass Index count with 9.63 and a number of correlating product species of 5029 closely followed by the highest Mass Index of run F, 8.67, but a lower mass species count of 316.

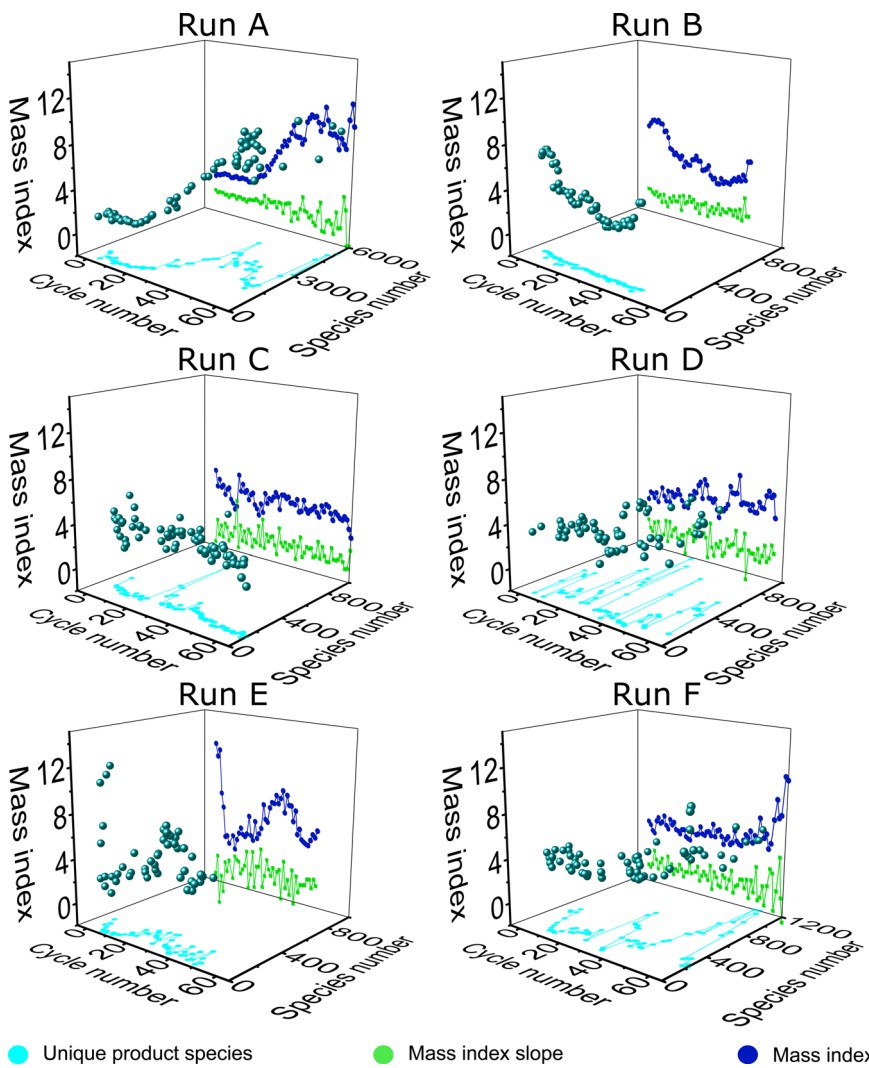

**Fig. 4 Selected experiments, which reached 65 cycles or more.** The light-blue line represents the number of unique product species versus the number of cycles, while the dark-blue line represents the calculated Mass Index for each respective cycle. The green line shows the corresponding slope of the Mass Index values. The spherical data points map these in 3D. Run A, B, C, and D had a cycle time of 6 h, while E and F had 3 h per cycle.

Run E has a highest Mass Index of 8.45 with 81 correlating species, the lowest number of product species found in this comparison. Run B has an index of 7.5 with a species number of 99 and run D has a Mass Index number of 7.05 with 70 product species. Run C has the smallest highest Mass Index number with 5.76 and 169 product species. These numbers show that the heaviest mass species in each cycle have a large effect on the calculated Mass Index, but the Mass Index enables the handling and interpretation of data sets with large numbers of unique species.

To understand how these complex product mixtures were being constrained by the recursive cycles, we decided to repeat previous runs by disabling the online decision-making aspect of the experiment and following previously recorded input trajectories. We wondered if a system that undergoes several equilibrium disturbances and many chemical changes could still be reproducible. To investigate this, we carried out a run as described previously (Fig. 3). The run started with a randomized input composition of 3 of the 18 input solutions, in this case acrylic acid, potassium pyrophosphate, and carbonyldiimidazole. After 10 cycles with this input composition, the 'Mass Index label' of each cycle was evaluated by the Python algorithm and the slope between cycles was calculated to be 0.034. As this was an increase,

the experimental parameters and the input composition were kept the same for the next cycle. From this point on, the slope was recalculated after each cycle and due to a continuing positive slope, the input composition (the starting material replenishment) was kept the same for a further six cycles. At cycle 17, the slope dropped below threshold ($-0.036$) and the input set was randomly changed to a new set of starting material solutions, ethyl acetate, formamide, and formaldehyde. This input composition was again evaluated up from its 10th cycle and kept up to cycle 42 after which the slope was $-0.015$ and in which the input composition was changed to ruthenium-(III)-chloride hydrate, pyridine, and copper-(II)-sulfate pentahydrate. This input set was kept until cycle 52 as the slope ($-0.229$) was below threshold after the first calculation with this input. The next input composition was resorcinol, pyridine, and formamide, which was like the composition before repeated for 10 cycles and the slope dropped to $-0.131$, until the last composition was randomly assigned, potassium pyrophosphate, ethyl acetate, and formaldehyde, until 65 cycles have been reached. With 65 cycles of 12 h, this means the experiment ran continuously for 780 h (32.5 days).

All input decisions were recorded and the exact same experiment with identical input compositions and cycle time of 6 h was repeated a further two times. Rather than allowing the algorithm

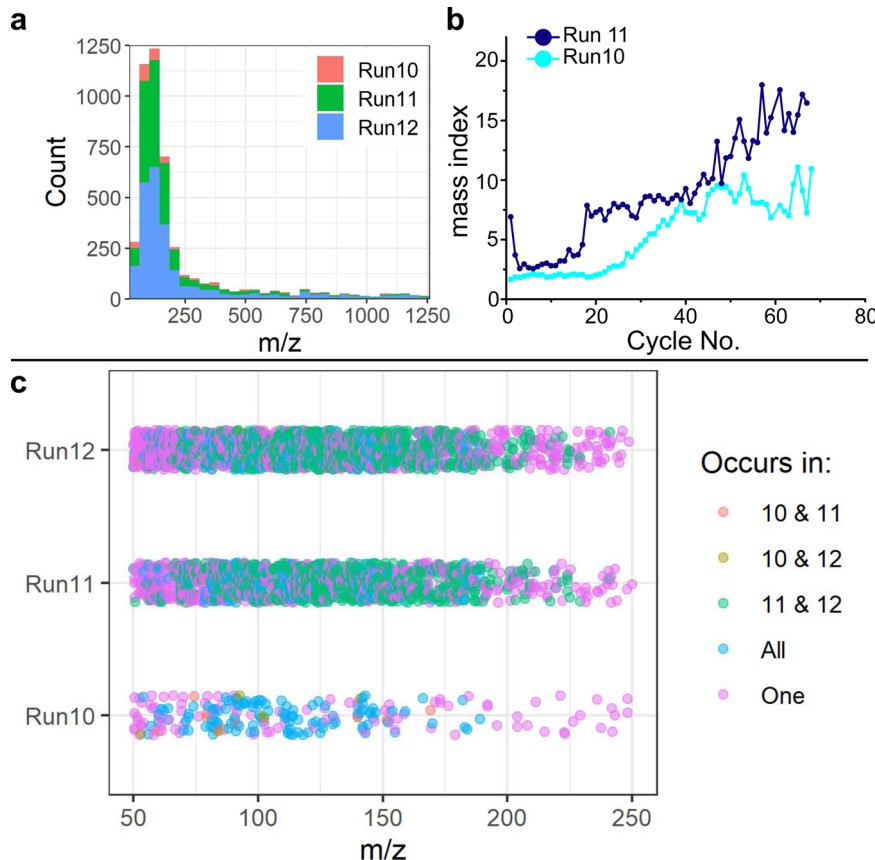

**Fig. 5 Comparison of reproduced identical experiments. a** Shows the m/z value distribution of individual experiments analyzed with the Thermo Orbitrap Fusion Lumos. Run 10 (green), Run 11(blue), and Run 12 (purple) are compared with all ions appearing in total in all samples after thresholding. The graph on the right (**b**) shows the online analytical result on the Advion L-CMS series of Run 10 and Run 11. The online analytical system failed at Run 12, which is the reason that there is not a third comparable dataset. **c** It shows a comparison of the offline analytical data of all three runs. The measurements of run 11 and 12 were performed at the same time, while run 10 was measured at a later date. This means that the differences between run 10 and the other runs may be partially due to a difference in performance.

to make decisions, these repeat experiments simply followed the same sequence of input parameters as the initial run. We can see that these experiments, prepared under the same conditions, differ from each other when measured online and as shown in (Fig. 5b), the curves of Run 10 and 11 lead to different curves of the mass index. To make sure the differences were not only from the Mass Index calculations, we analyzed and compared all three runs, based on the raw MS data and the HPLC-DAD data as well. We further tested all collected samples from each run on a more sensitive MS instrument after all runs had been completed. Those results (Fig. 5c) show that many of the products from these runs were identical but also differences in the overall product distribution of the different runs, especially in higher mass ranges, where we see many features that are unique to each run.

Differences in the reproduced experiments are observed, even when an automated system was used for the preparation of the experiment. Figure 5c reveals that many features with high m/z are unique to one run and that more such features are seen in the repeat runs. Given how the Mass Index is calculated, with especially heavy masses being weighted higher, this helps explain the origin of differences seen in the repeat runs compared with the initial experiment. Even with minerals used from the exact same origin, each grain size selected and washed prior to the experimental start, the mineral environment will not be identical and the resulting surface chemistry and the adsorption effects will lead to a small variation. In addition to this, each MS run, even of the identical sample, will change slightly through ionization and can

lead to a slight variation in the overall detected product species of the run. With an experiment based on several repeated cycles and a comparison of each species occurring, small variations, especially in the higher mass features, can lead to the observed differences in the reproduced runs. During all experiments described above, the Mass Index algorithm in Fig. 3, was used as the decision-making metric. It is used as a tool to explore unconstrained multicomponent systems algorithmically and to approach the current analytical difficulties in the field. Complex chemical mixtures with thousands, or tens of thousands, of different product species are often seen as an analytical problem, and the conventional approach of analyzing every single product species is not feasible. This is why researchers have attempted to develop a more system-level approach to detect changes and trends in spectra instead of attempting to identify isolated species[3,20,21]. The Mass Index is not able to capture the complexity of the data of a cycle completely, but rather simplifies the system. Previous recursive chemistry experiments already raised the problem of addressing every single feature of an experiment, but as the number of experimental cycles increases, the need to effective heuristics and metrics becomes even more significant[3,20,21]. The Mass Index value is a simple metric for each cycle, enabling a fast, algorithmically driven comparison and the adjustment of the experiment in real time based on these data. It could be argued that the algorithm is too simple and cuts out too much of the data. To investigate this, other algorithms looking into information entropy or the weight by intensity values have

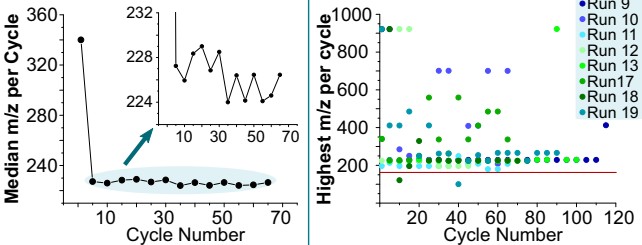

**Fig. 6 Experiments compared by the highest m/z value.** On the left, we present a diagram, which shows the median of the highest m/z value of each experiment compared with the cycle number. As the first value is very high, a zoomed-in version of the plot is shown as an inset. On the right, we show the highest m/z value over the threshold of each cycle of selected runs. The line (red) in the plot represents the m/z value of the heaviest input solution.

been tested (detailed description of each algorithm can be found in the SI). We observed that the algorithms tracked with each other, as for example, in run E cycle 120 (Fig. 4); the Mass Index algorithm and the mass-by-intensity algorithm values increased while the information entropy value dropped. This indicates a drop in unique species while the m/z values increased. While this showed that the algorithms worked in general, we have not been able to get further insights into the actual data by using different algorithms. In addition to this, the Mass Index needs to be simple, not just to keep human bias low, but also to enable a calculation in a short amount of time, as the analyzed sample is compared immediately in order to make a decision regarding the input composition of the next cycle.

The next step was to look at the samples systematically offline to compare different chemical systems: different runs had the same principle and method, but with 18 input solutions available, the chemical compositions, cycle, and run times differed considerably. We compared the highest mass over charge ratio between cycles and between all runs of comparable cycle length. As shown in Fig. 6, there is no average increase in mass to observe with increasing cycle number. This is consistent with the Mass Index observation, as already that data showed differences. The experiments with a high m/z ratio in Fig. 6 are correlated with the highest Mass Index runs.

When looking at Fig. 6, our first observation is the high m/z ratio of the first cycle, as this seems to occur in several cycles. We believe that this observation can be related to the use of minerals. The minerals we used were new and washed for each run individually, so we do not think contamination to be the cause based on our mineral controls (SI section 4.1.3); however, some leaching of elements would be possible in the first five cycles (SI section 2.2). However, the newly washed mineral surface could be adsorbing some species, leading to a lower concentration of products, and alternatively, the clean mineral surface itself could be catalyzing the breakdown of bigger molecules. In either case, the phenomenon is limited to the first cycle as the surface becomes covered by material from the experiment. For further understanding of how the mineral interacts in this system, the analysis of the mineral surface through electron microscopy or MALDI–TOF–MS would be an interesting future extension of the work. The median of all m/z values of all cycles is under 240 m/z, while the individual cycles on the right-hand side of the figure show a very distributed scattering. Most of the highest m/z species are heavier than the heaviest starting material m/z, which is marked with the red line. Even with no direct observable pattern in the scattering of the plot, there is an area around 240 m/z, which seems to have the most abundant products of the cycles. Taking into account that the input reagents and compositions in

all these experiments and cycles have been very different, it is an interesting observation that most of the highest m/z species lay in the same m/z area. Of course, it should be noted here that species with smaller masses usually fly better in the mass spectrometer and are therefore easier to detect. When comparing these findings with the Mass Index behavior shown in Fig. 4, we see that the Mass Index can increase even when there is not a dramatic increase of the mass of the heaviest product in the cycle. The highest observed Mass Index was in run A with 9.63 and 5029 unique product species. This is interesting as that amount of product species is on the higher range of the observed count. This means that in this particular cycle, the mass of the heaviest product was so high that the Mass Index was calculated high even with that number of species. This can lead to the conclusion that in this cycle, the randomly chosen input set leads to an increased complexity of the product mixture.

In conclusion, we present a fully automated, algorithmically controlled platform for prebiotic chemistry experiments. The long-term experiments run longer than 30 days and have been capable of executing up to 150 consecutive recursive cycles with different chemical compositions, stirred and heated in the presence of a mineral environment. The automated system was tested using a simple heuristic analysis of complex mixtures based on a system-level perspective, rather than focusing on a narrow set of 'prebiotically plausible,' or 'biologically relevant' substrates. The reactivity of the product library was simulated, showing that the selected reagents are prone to a product mixture complexification. This enabled the use of an algorithmic approach to the problem rather than an approach driven by narratives, by directly feeding back knowledge of the current experimental cycle for the next one. The algorithm controlled and adjusted the experimental conditions (composition of the feedstock) on the fly by using an automated decision-making metric, which enabled the computer to interpret the MS data and make conclusions using data from previously executed cycles. Thanks to this feature, this system could be used by other scientists to explore the expansion of the chemical reaction network starting from simple organic compounds to include more complex molecules, adapting the feedstock based on the identification of key chemical species. Such an experiment could be used to test long-standing hypotheses about the emergence of biochemical pathways before enzymes[11,12].

The algorithm used here presented a first step to approaching the problem of analyzing messy chemical systems and on top of that, enables the experiment to continue without a human-made decision. The data show that the algorithm succeeds in controlling the experiment, leading to different behaviors of each experiment while reducing the chemical mess to the total number of possibilities and avoiding a combinatorial explosion. The data presented showed unconstrained multicomponent systems, their behavior, the borders of their reproducibility, and that most of the heaviest species produced are in the zone of a mass around 240 m/z. The Mass Index, which generates a relation between the heaviest product species and the number of species, enables the handling of large data sets. We achieved the performance of recursive experiments generating up to 5256 unique product species in a single cycle (cycle 65 of run A in Fig. 4), while the correlating Mass Index was 9.12. In the same experiment, the cycle with the highest Mass Index 9.63 was found with 5029 unique product species detected (cycle 63 of run A in Fig. 4). Automation can only go as far as theory; writing an algorithm for an experiment of an unknown chemical system, without knowing the outcome of the experiment, appears to result in various complicated problems. Further development is needed to algorithmically explore phenomena in complex mixtures, but these results present an important first step to autonomous unbiased Origin of Life experiments and open the door for exciting future

research. Therefore, we hope that others will adopt our approach described here, and in the SI, so a common experimental standard can be adopted for these types of experiments. We hope this will enable the development of a global effort for 'big-data' origins-of-life search experiments. A platform, like the one described here, could be used to search for increasing complexity by which simple molecules become complex chemical systems. Indeed, we were able to show how new species were able to persist over time and not be diluted away, and this could be due to a range of processes, including molecular replication, amplification for example. In future work will will use assembly theory and mass spectrometry to follow the assembly of complex molecules under prebiotic conditions to explore the mechanism by which selection can operate outside of biological systems[31]. Identifying those processes that lead to the robust complexification of chemical mixtures could one day lead to the missing link for the chemical-to-biological transition.

## Methods

**Experimental details**. All materials and solvents were purchased from commercial sources (Sigma Aldrich, Fischer Scientific, and Alfa Aesar), unless otherwise stated, and used without any further purification.

Ulexite and quartz were obtained from Richard Tayler Minerals, Cobham, Surrey, England, and crushed in a Mad Mining Rock Crusher with a Solid Steel Frit.

**Mineral wash workflow**. An equal mixture of 2 g of quartz ($SiO_2$), ulexite ($NaCaB_5O_6(OH)_6\cdot5H_2O$), and pyrite ($FeS_2$), each was added to the reactor. All minerals used have been sieved to a size between 2 and 4.75 mm. The minerals have been boiled and stirred in HPLC-grade water for 2 h and continuously rinsed with fresh HPLC-grade water, until the solution in touch with the minerals remained clear. After that, the minerals have been dried and directly transferred to the reactor.

**Chemical input preparation**. All input solutions have been prepared on demand in HPLC-grade water as follows:

All chemical inputs have been used as a 0.1 M solution, the only exception was ruthenium-(III)-chloride hydrate that was used as a 0.01 M solution.

**Mobile-phase preparation**. For platform HPLC analysis, 0.1% formic acid was added to HPLC-grade water or HPLC-grade acetonitrile and the solution was sonicated for an hour before being set up on the instrument. For high-resolution HPLC–MS, the procedure was similar, but LC–MS-grade water and acetonitrile have been used.

**Platform high-pressure liquid chromatography (HPLC–DAD)**. Gradient HPLC analysis was performed on an Agilent 1260 Series (Agilent Technologies) instrument equipped with a quaternary pump (G1311B) and a diode array detector (DAD) (G1315D). The sample was injected from the sample loop on an Agilent Infinity Lab Poroshell 120 Eclipse EC-C18 UHPLC Guard 3.0 × 5-mm guard column that was connected to an Agilent Poroshell 120, 120 EC-C18, 4.6 × 150-mm column, kept in a column compartment (G1316A) with a controlled temperature of 30 °C. The method used was a gradient method with 95% 0.1% formic acid added to HPLC-grade water and 5% 0.1% formic acid added to HPLC-grade acetonitrile (MeCN). Over 10 min, the organic (MeCN) flow was increased to 40%, after another 5 min it was at 50% MeCN. After 20 min, a flow of 100% organic mobile phase was reached. After that, the mobile phase was switched back to the initial 95% water and 5% acetonitrile, and a 20-min flow was maintained for column cleaning. The flow rate through the whole run was 0.5 mL/min, while 0.2 ml/min flow was maintained between runs. Elution was detected by UV ($\lambda = 200$, 215, 245, and 300), and samples were run for 40 min in total. The performance of the HPLC column was checked with a caffeine standard solution, based on the directions in DIN 20481, on a regular basis.

**Benchtop electrospray ionization mass spectrometry (ESI-MS)**. The Benchtop ESI-MS was used for all online measurements during the platform experiments and for the Mass Index calculation. Data are presented in the paper in Figs. 4–6 and in the SI in Figs. 4–6, 10, 18 + 19. The ESI-MS analysis was performed on a Benchtop 'expression L-CMS' system from Advion. After the sample passed the HPLC-DAD, it went through a split valve resulting into a flow of 0.2 mL/min injected into the ESI system. The mass spec was run in positive and negative mode, switching between both modes during analysis with a switching speed of 50 ms. The positive mode turned out to be more useful and only the results of this mode were used for the algorithm. The m/z range was set from 10 to 2000 m/z and the scan time was

3345 ms at a scan speed of 595 m/z/sec. The ion source parameters were as follows: capillary temperature 300 V, capillary voltage 120 V, source voltage offset 20 V, source voltage span 30 V, and the source gas temperature 200 °C. All settings were the same in both positive and negative modes, except for the ESI voltage, which was 3500 V for the positive mode and 2500 V in the negative mode. A calibration was performed regularly with a MS tuning mix (Agilent). Data were analyzed using the Advion Data Express software.

**Electrospray-ionization mass spectrometry (UPLC–ESI-MS)**. The Orbitrap ESI-MS was used for further offline analysis. Data are shown in the paper in Fig. 5. The sample was run through a Thermo Vanquish UHPLC system and injected 10 μL on an Agilent poroshell C18 2.7 um 4.6 × 150 mm column with a flow rate of 0.5 mL/min. The column temperature was maintained at 40 °C and sample vials kept at 50 °C. Mobile phases were 0.1% formic acid added to LC–MS-grade water and 0.1% formic acid added to LC–MS-grade acetonitrile (MeCN). A gradient was applied, starting with 1-min equilibration time, the run started with 95% aqueous phase. Over 40 min, the organic (MeCN) flow was increased to 20%, after another 20 min, it was at 60% MeCN. After 60.5-min runtime, a flow of 95% organic mobile phase was reached. After 70.1-min runtime, the mobile phase was switched back to the initial 95% water. The flow rate through the whole run was 0.4 mL/min and the total runtime was 75 min. The chromatographic separation was then ionized in a HESI ion source with 40-psi gas and a +3.4-kV voltage applied to introduce the sample into the mass spectrometer (Thermo Fusion Lumos).The acquisition was run in positive mode and the system was calibrated before each run. The mass range was from 50 to 1000 m/z. The fragment ions were analyzed in the Orbitrap with HCD fragmentation set at 35%. The isolation window was set at 1 Da and the resolution of the MS1 scan was 120,000.

## Data availability
All data necessary to evaluate the conclusion of this work can be found in the Supplementary Information or Supplementary Data 1. Due to the quantity of the produced data, the full raw data are only available on request.

## Code availability
The code used in this work is provided in Supplementary Data 1.

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

## Acknowledgements

We would like to thank J. Szymański for writing the decision-making algorithm, L. Wilbraham for help with the Molecular Transformer, E. Carrick for help with the offline analytical workflow, and J. McIver for technical support. The authors gratefully acknowledge financial support from the John Templeton Foundation (Grant 60025), EPSRC (Grant Nos EP/L023652/1, EP/R01308X/1, EP/J015156/1, EP/P00153X/1), the Breakthrough Prize Foundation and NASA (Agnostic Biosignatures award #80NSSC18K1140), and ERC (project 670467 SMART-POM).

## Author contributions

L. C. devised the concept and the initial algorithm and platform including the pseudo code to run the system. The platform was built by S. A., G. J. T. C. and G. K. wrote the software to operate the system. C. M. built the network simulation. S. A. performed the experiments and analyzed the data. S. A., G. J. T. C., C. M., and L. C. wrote the paper.

## Competing interests

The authors declare no competing interests.
