## [Peer Review File · Nature Communications]

Reviewers' Comments:

Reviewer #1:

Remarks to the Author:

The authors present an autonomous robot to run experiments running in several rounds over several weeks in an attempt to study "artificial life experiments". Their platform is a nice setup to study reactions running in multiple cycles. The setup and experimental details are very well documented.

The idea to use it to reproduce prebiotic chemistry is a good one, but the experiments and the conclusions fail to convince me (see "Main concerns" below).

I think that the authors are attempting too much at once, without taking the time to evaluate the single decisions taken for the experiment setup; in particular:

- the choice of the initial conditions (input molecules and minerals)
- the choice of the metrics
- the iterative setup of the experiment (dilution problematic below)
- the decisions made during the iterative setup

Main concerns:

- Dilution. The authors explain that at each cycle, 30% of the volume is kept from the previous cycle. This leads to an exponential decrease of the concentration of added compounds in the reaction mixture. Already after 10 cycles, the relative concentration is $0.3^{10} = 6 \cdot 10^{-6}$ compared to when a compound is added. Considering (from the SI) that on average the added compounds are changed every 10 cycles or so, it means that we can basically consider only the three compounds being added in the current window and in the previous window: everything from previous windows (reactants and products) will be present in negligible concentrations and not likely to be above the threshold for the calculation of the mass index. Therefore, if I see it correctly, the experiment is rather about a "moving window" of reactivity, where the observations will mainly relate to reactants being added in neighboring cycles. Considering this, for the observations the total number of cycles is irrelevant: even if there are 150 cycles, the observations at the 150th cycle will only "see" things that were added in the last ~ 10 cycles. To me, this is a major shortcoming and puts a big question mark on the conclusions of this paper, as it prevents any real build-up of molecular complexity.

- Mass index. I find this metric very questionable (at least, the paper does not convince me of its usefulness). First, because of the dilution issue mentioned above, one cannot expect the global trend of mass index to be meaningful except in the first ~ 10 -20 cycles, as the initial products will have virtually disappeared. Second, the paper does not convince me that this metric is actually a good descriptor of what is happening in the experiment. I do understand the need for simple observables when observing mixtures, but it is difficult to take this metric seriously without explicit demonstration or proof of the usefulness based on an actual analysis of the products. Without this, I can only look at the formula, and notice that it is, in my opinion, not very robust: a little change in the MS intensity (either from one cycle to the next, or because of experimental noise) can lead to one species being above the threshold or not. Such a little change in intensity can therefore lead to a big jump in the value of the mass index because the mass range or the number of considered peaks will suddenly be different. This may be one of the reasons why the reproducibility test failed.

- High initial values of m/z . The article is about the build-up of molecular complexity. There, anyone would expect the highest m/z value as a function of the cycle to be increasing ("looking for an increase in the mass of the product species"). The initially very high values of m/z , however, prevent any subsequent conclusion: how can one believe any of the following m/z values (or mass index values) if the initial solution is contaminated with complex molecules already? This makes me question all the subsequent results (including the mass index). The authors guess that this may be related to the added minerals. In this case, why not try the same experiment without those minerals?

- Number of product species. I am struggling to understand how there can be more than 20000 product species starting from only three compounds, and how one can give any meaning to this value.

- Combined with the previous concerns, I find the lack of any actual analysis of the content of the reaction mixtures, in terms of chemical structures, very disconcerting. This would actually be a very interesting piece of information! Especially if one sets out to reproduce reactions leading to

life, anyone would be curious to see what is happening, even if only a few examples were given. The abstract even says "explore unconstrained multicomponent reactions" - I do not consider the analysis of a single metric to be an actual exploration. Only measuring an increase in complexity is not as exciting and not as unexpected.

- I would also find some basic checks in terms of reactions and chemical structures reassuring, given all the points above: it could at least show that the system is working as expected, and for instance give further insight about the large initial m/z values or the peaks at 240, which are not explained. It would also show that the authors actually understand the chemistry that is happening.

- Reproducibility. The failed attempt at reproducibility is worrying. It was even set as one of the goals, "to do this in a reproducible way". But the data shows that this is not the case: the curve with the mass index is significantly different!

Other general comments:

- I am not sure what to think about the conclusion on the ability of the system to do "long term reactions" (in the title!). Because of the dilution issue mentioned above, I am not sure that one can consider this to be correct. At most, it shows that the software and hardware integration is robust. Also, when reading the title, I was expecting "long" to mean several months or years.

- How were the 18 starting mixtures chosen? The authors write a full paragraph that sounds a bit defensive about them being chosen by humans, but I would be very interested to know why these 18 specific compounds were chosen.

- I find that the paper over-emphasizes the "origin of life" aspect of the work. The choice of the 18 starting mixtures is not justified, and no single product molecule is given.

- In the abstract: "We show that the robot can discover the production of high complexity molecules": there is no proof of any high complexity molecules; the paper does not contain a single chemical structure except the 18 starting compounds.

- In the abstract: "experiments that allow testable hypotheses for the emergence of life from prebiotic chemistry": I don't see any such hypothesis in the paper, and it is not explained how they would be testable, even if the system allowed for reproducible experiments.

- "we first explored the chemical space accessible to the 18 input reagents computationally". Such a good idea, but what a pity that the authors do nothing more with it than seeing that there are 2206 possible reactions.

- At the end of each cycle, 70% of the solution is stored. The experiments consist in 60-150 cycles, but on the picture I see only ~20 vials for storing, which means that either they were changed by hand, or not all the cycles were then stored.

- "not only automate a vast number of experiments, but also make decisions on the fly about which routes to follow". It would be great to inspect other ways to make decisions than to rely only on the mass index - and I find the decision based on the slope of the mass index somewhat arbitrary.

- "The data shows that the algorithm succeeds in controlling the experiment, leading to different behaviors of each experiment while not generating the chemical mess, the simulation showed." The authors do not explain how the chemical mess is avoided - this is not visible in the data. Also, if the authors base this statement on the computational approach, they should take in consideration that in their experiments, only a subset of the product molecules will be possible at any time (dilution issue above), and that several products may have the same chemical formula and therefore lead to a single peak in the MS spectrum.

Details:

- in the abstract: the size of the data (20 GB) is irrelevant.

- Why did the authors do the reactions under a nitrogen atmosphere? Was it a deliberate choice?

- "just 3.7% of all experiments reported to Reaxys": considering the size of Reaxys, this is a lot of reactions! I wouldn't take this as an argument that there are few long reactions reported in chemistry.

- Figure 1 would be more readable if it was larger.

- Page 7: "known to be too complex to be analyzed". I would expect further clarification at this point.

- "Taking in account how the index is calculated and that especially heavy masses are weighted higher, the data is complementing itself" What does this mean?

- "It could be argued that the algorithm is too simple and cutting too much of the actual data, but when testing alternative algorithms, we observed that all of them have been complementing each other" What other algorithms? What do you mean by "complementing each other"?
- "as well to enable a calculation in a short amount of time": this is not a good argument. Computers can analyze huge amounts of data in seconds!! This is nothing compared to the weeks of the experiment.

Typos and languages:

- everywhere: please review the language. I understand that the first author is not a native speaker and I would recommend a native speaker to do extensive proof-reading. Many formulations are strange. Also, a lot of commas are placed where not necessary in English (f.i. page 12 "There are several reasons, why differences").
- p. 3: "now only" -> "not only"
- p. 3: "a heterogenous reaction environments" -> remove "a"
- p. 5: "4 syringe" -> "4 syringes"
- p. 5: "purpose- build" -> "purpose-built"?
- p. 6: "Everything was stirred and heated until the stirring was terminated one hour before the sampling was due" this is not very understandable
- p. 5: "which choose" -> "which chose" / "which chooses"
- p. 8: "bespoke" looks strange in this context
- p. 8: "By calculating this number, we are able to use it as a metric, which enables the comparison of cycles with each other, fully automated": please reformulate
- p. 8: "randomized" -> "randomly"
- p. 8: "As well as the index can" -> "Furthermore, the index can"
- p. 8: "Experimental data, of the mass index, their slope" what does this mean?
- p. 8: What is a "hill climber rise"?
- p. 10: "We wonder" -> "We wondered"
- p. 11: "We can see that these experiments...": long sentence that is not very readable
- p. 15: "The reactivity of the product library was simulated, and the reagent network showed, the reagent library was prone to create a messy product mixture, combined under the used conditions" not very understandable
- p. 16: "HPLC-grad" -> "HPLC-grade"

Reviewer #2:

Remarks to the Author:

Cronin and co-workers describe an automated laboratory system for long-term experiments setup to explore unconstrained multicomponent reactions. Their system's acquisition of data from mass spectrometric, ability to adjust reagent input based on such data and user-defined algorithms allow these researchers to explore the immense chemical space of products that is possible with the input of eighteen, and possibly more, organic compounds and minerals. Results are provided from runs that were carried out for up to 4 weeks with 150 algorithmic decision cycles. These experiments provide validation that their automated system, as intended, is able to generate and deal with the extremely large data sets that result from recursive-unconstrained chemical reactions.

Overall impression:

The authors state that running long-term, and potentially open-ended, experiments is an approach to origins of life, prebiotic chemistry, and artificial life research that is necessary but currently lacking from these fields. I agree. These fields need such approaches that break from the more standard approaches of synthetic organic chemists and prebiotic chemists who have traditionally sought to produce individual compounds through carefully designed and controlled reactions. In contrast, the approach being advanced by the Cronin and coworkers minimizes human bias to the selection of reagents and reaction conditions. After that, the chemistry of the system is allowed to follow its own path, with some direction provided by a relatively simple metric, such as m/z distribution.

While I appreciate the motivation behind the work described in this manuscript and the potential for the automated system developed to reveal chemical transformation and synthetic routes that might otherwise never be revealed, actual chemical results presented in this manuscript remain rather obscure. The data provided show that recursive experiments guided by autonomous computer control can lead to long term increases in mass index, for example. However, without a revelation and discussion of the actual molecules that are produced from these processes it is not possible to appreciate how interesting or unexpected the chemistry is that emerges from this non-rational approach. In this context, I feel that this manuscript is better suited for publication in Nature Methods or Review of Scientific Instrumentation.

Specific issues to address:

The word "prebiotic" should be removed from the title, are at least qualified as a potential application of the instrument profiled. The authors note several times that it is not their intention to show prebiotic chemistry in this work, e.g. in the conclusion: "The automated system was tested using a simple heuristic analysis of complex mixtures based on a system level prospective, rather than focusing on a narrow set of "prebiotic plausible," or "biologically relevant substrates". Also on page 6 "We tried to concentrate on the function of the building blocks itself instead of taking prebiotic chemistry narrative in account (No "common" autocatalytic cycle precursors, sugars or amino acids)." Similar statements are made in the SI. The work may have potential future applications relating to abiotic chemistry, but no such arguments or data are presented in this work.

There are several sentence fragments, typographical errors and use of colloquial language throughout the body of the text. As two examples, the first line of the Results reads "In this work we set out to design a system that can now [not?] only automate" and on page 8, it is stated "experiments presented follow some point a hill climber rise"

Strictly speaking, the term "mineral" refers to a natural substance. Is that the case for all of the inorganic salts that are referred to as minerals. If not the point should be made that the minerals used are synthetic. Otherwise further confirmation of mineral purity may be necessary to guard against natural substance heterogeneity contributing to experimental variations.

Is the pH of the reactions measured or controlled? This information could be important, as it may alter the kinetics of potential reactions, and possibly the mineral solubilities (altering the effects of potential "surface" chemistry).

Were the minerals themselves replenished after each cycle? What controls were there for this? It seems like continual addition would lead to a build-up of solid material in the reaction vessel. The authors state that further analysis on the effects of minerals could be the subject of future work, but more information about how the mineral content was controlled throughout this study is still needed.

The authors state that the cycle times vary from "3 to 12 hours", but no information is given about the duration of cycles during any individual runs. This duration information is important for reproducibility of the results. For example, on page 10, all of the input changes are laid out for a given experiment (summarized in Figure 5) that was repeated 3 times, but information on the cycle time is not provided. I would assume that cycle times are the same, but these need to be stated explicitly. Also, the times required for HPLC-MS runs could be important if this analysis must be completed before the next cycle/ input change is allowed to proceed.

For Figure 4 what is the significance, if any, of the spherical data points in the 3D plot?

Figure 5: The figure caption labels refer to the incorrect positions in the image (right/left etc.)

Figure 5: Were runs 10, 11 and 12 identical replicates?

Figure 5: Are the linear plots suppose to show that there is a correlation perfect or is a straight line fully expected due to the nature of the data being plotted? If the latter, different presentations

of this data should be used so as to emphasize what is interesting in the data being compared.

Page 12: As a possible explanation for the different result obtained for replicates of the same experiment, the authors state: "In addition to this, each MS run, even of the identical sample, will change slightly through ionization and can lead to a slight variation in the overall detected product species of the run." Can a reference be provided for data that supports this claim (perhaps using standards). It would seem that if the solutions had identical concentrations of compounds, their ionizations would not differ. Also, variances in mineral composition are mentioned as a possible reason for non-reproducibility. This argument could be likewise be fortified with references or control experiments.

Page 13: The authors state "It could be argued that the algorithm is too simple and cutting too much of the actual data, but when testing alternative algorithms, we observed that all of them have been complementing each other, but have not been able to get further insides about the actual data" What other algorithms were tested?

In the Conclusions the authors state "The data shows that the algorithm succeeds in controlling the experiment, leading to different behaviors of each experiment while not generating the chemical mess" Do the authors mean to say "the chemical mess is reduced when compared to the total number of possibilities"? Could this also be partially due to the 106 cut-off imposed during mass index calculation? If so, this should be acknowledged.

Page 19: pressure units/or gas flow rate units missing

SI Figure S5 and S6: Sulfate anion (?) structure is incorrect.

SI Figure S8: There are two Resorcinol chromatograms. Is one mislabeled? Additionally, the top Resorcinol chromatogram look identical to the pyruvic acid chromatogram.

Reviewer #3:

Remarks to the Author:

Gaining insight into navigating complex chemical networks and searching for a functional increase of complexity is of utmost importance in modern prebiotic chemistry which takes a step beyond the classical synthetic organic chemistry approach based on individual reactions. This paper by Asche et al presents an automated platform that allows for multicomponent reactions to be run over long periods of time and in a large number of substrate addition-analysis cycles. The outcome of each cycle is monitored by HPLC-MS, and the obtained MS data are fed to an algorithm which can randomly alter the substrate input based on how much the mass of the product species increases.

The premise of the paper is extremely interesting, especially in light of the gradually shifting paradigm in prebiotic chemistry, which now begins to notice the importance of reaction networks and systems chemistry over the classical stepwise syntheses. However, despite the very interesting context, this paper leaves me somewhat confused. One thing I am missing here is an explicit statement why this study was undertaken. Even if it was something along the lines of "we wanted to mix random chemicals to see what happens" (which I do not believe was the case), this should be stated already in the abstract (i.e. what is the existing problem and why it needs solving). Otherwise, as a first-time reader, I have no idea what the authors are trying to tell me and what the implications are. Even the authors' own previous work is not directly used to frame the discussion (it is hidden in the references)! Secondly, stating the working hypothesis is just as important because without it the manuscript is a collection of data, without a clear storyline.

As if referring to the chemistry of the experiments presented here, the language of the manuscript is rather messy in places. Often, the manuscript contains compound sentences with multiple levels of clauses. This leads to two problems. One is that sometimes a preposition or a verb might be missing here and there, and the sentence sounds quite bizarre. Another is that much of the paper's message becomes lost—and this problem becomes particularly frustrating in the abstract, which is the weakest part of this paper writing-wise. Still, I believe this can be easily fixed.

Another problem is that the paper still requires some work in terms of phrasing the story and

explaining the premise of the study in a way accessible to a general chemist. Many details are missing, some important information is only mentioned briefly (e.g. the choice of reagents/catalysts), but the context/rationale behind it is left to the imagination, or explained in a different section (with no reference to that section). If certain data or details or procedures are presented in the SI, references to appropriate Tables/Figures are needed in the main text. Finally, sometimes a claim is made but not supported with a reference to the literature. I will list all these issues below.

Specific comments and suggested corrections – in a chronological order:

- Page 2 (top): “massively parallel experiments” – I am not sure what this means. A large number of experiments performed in parallel?
- Page 2: “life was thought to have emerged ca. 3.8 B years ago” – a reference is needed
- Page 2: “Previous approaches to prebiotic chemistry, which have their origin in synthetic organic chemistry, intentionally try to limit the accessible size of the chemical space in experiments” – while in principle I agree with the authors’ sentiment here, limiting the size of the chemical space does not always have to be a bad thing. If the focus of a study is to investigate a mechanism or a certain reaction on a certain functionality, it only makes practical sense to limit the number of variables (or—to “minimise the noise”). Whether the said reaction/mechanism is texted in a broader context afterwards is, of course, another story, and I agree the synthetic organic chemistry school in prebiotic chemistry do generally refrain from this.
- Page 2 (bottom) – In this context I suggest also citing the following:
 - Vincent et al. Chemical Ecosystem Selection on Mineral Surfaces Reveals Long-Term Dynamics Consistent with the Spontaneous Emergence of Mutual Catalysis. *Life* 9, 80 (2019).
 - Baum, D. A. The origin and early evolution of life in chemical composition space. *J Theor Biol* 456, 295–304 (2018).
- also current ref. 4 seems suitable to be mentioned here as well.
- Page 3 – “a system that can now only” – should be “not only”
- Page 3 & 4 (Chemical space) – an explanation is needed why these molecules were chosen and not others, and what a “possible reaction” means in this study. Thermodynamically possible? The rationale behind the choice of molecules is indeed hinted at on page 6/7 in the “Data from the platform” section, but in my view this is far too late. I suggest describing the choice of reagents early in the “Chemical space” section.
- Page 4 (top) – “any combination that gave a score greater than 0.8” – to make this paper accessible to a general chemist, a brief explanation of how this works and what this value means is due here. Otherwise this number is meaningless.
- Page 4 – Figure 1 – it is impossible to see what the molecules are even at 400% zoom. Consider resizing.
- Page 5/6 – heterogeneous mixtures and minerals are mentioned, but there is no reference to any table/list of these heterogeneous substances/minerals. How were they chosen?
- Page 6 – the sampling is limited to whatever is dissolved in the aqueous solution, but is there any indication of products adsorbed on the mineral surface? It would be a good idea to subject the minerals to surface analysis. Analysis by MALDI-TOF-MS is suggested as future work on page 14, but based on how many times the minerals are mentioned in the paper, at least a qualitative XPS/XRD would already be very informative.
- Page 6/7 – “We tried to concentrate on the function of the building blocks” – what function exactly? How/why was this function selected? Could this be elaborated on?
- Page 7 – “a chemical system known to be too complex to be analyzed” – a further comment is needed or at least some references to back up what is “known”. Perhaps these (and refs therein)?
 - Schmitt-Kopplin, P. et al. Systems chemical analytics: introduction to the challenges of chemical complexity analysis. *Faraday Discuss* 218, 9–28 (2019).
 - Geisberger, T. et al. Evolutionary Steps in the Analytics of Primordial Metabolic Evolution. *Life* 9, 50 (2019).
- Page 8 – the meaning of different Mass Index values with respect to the increasing/decreasing product complexity and the number of product species is described in a rather confusing way. Presenting general rules in a tabularised form might be helpful. Commas in random places sometimes change the meaning of sentences—please check this.
- Page 9 – The Mass Index values, as well as the number of product species in each run could be added to Figure 4. Presenting the graphs in 2 columns and 3 rows, with these values added, instead of the current 3 columns x 2 rows, would make Figure 4 much more accessible to the reader.

- Page 10 – how I understand it, the number of cycles the system is subjected to after the addition of the starting materials is not a set value, and depends on whether the slope falls below the threshold or not. Is this correct? It is not immediately obvious from the way it is described.
 - Page 12 – “With minerals used from the exact same origin” – it is already page 12, the minerals have been mentioned several times and still no mention of what they actually are
 - Page 12 – “With an experiment based on hundreds of repeated cycles” – this is not really the case here, is it?
 - Page 12 – “Complex chemical mixtures with several different product species are increasingly leading to an analytical problem and the well-established way of analyzing every single product species of a MS spectrum is not feasible” – analysing “several different product species” by MS methods is definitely feasible. Please rephrase.
 - Page 12 – “This is the reason that researchers attempt to develop a more system level approach, to look into changes, functions and trends in spectra instead of attempting to identify isolated product species” – if there are indeed other attempts, references are needed. I am sure this is known to the authors but to analyse a complex mixture one need not isolate the species – metabolomics approaches fare quite well here (e.g. Keller, M. A., Turchyn, A. V. & Ralser, M. Non-enzymatic glycolysis and pentose phosphate pathway-like reactions in a plausible Archean ocean. *Mol Syst Biol* 10, (2014).)
 - Page 12/13 – “Previous recursive chemistry experiments already raised the problem of addressing every single feature of an experiment” – references to these previous experiments are needed
 - Page 13 – “In our experiment, we are aiming for a rise in the complexity or information content of our input molecules, looking for an increase in the mass of the product species in addition to the use of the algorithm.” – this is a great way to state the aim of this study but it should appear on page 1, and not page 13.
 - Page 14 – “This observation can be related to the use of minerals. The minerals we used have been new and washed for each run individually” – the fact the minerals were used fresh and prewashed has been stated several times in the text already, but the identity of these minerals is (still!) unknown
 - Page 15 – “platform for artificial life experiments” – throughout the paper I was convinced the application of the platform was in prebiotic chemistry/origin of life experiments. Artificial life is quite a different domain, and it cannot be used interchangeably with prebiotic chemistry. One could compare the origin of life research, conceptually, to an interpolation between geochemistry and biology, while artificial life is essentially an extrapolation.
 - Page 15 (bottom) – “while not generating the chemical mess” – any idea why? In an “unconstrained multicomponent system”, what provides this constraint?
 - Page 15 (bottom)/Page 16 (top) – “most of the heaviest species produced are in the zone of a mass around 240 m/z”, “We achieved the performance of recursive experiments generating up to 28602 unique product species in a single cycle” – what does this mean for the bigger picture? What are the implications?
 - Page 17 – Mineral wash workflow – again, what are these minerals? In fact, I cannot find this information in the SI either. Minerals like FeS₂, MnO₂ or SiO₂ (off the top of my head) have strikingly different properties and could dramatically change the chemical landscape in some cases.
 - SI – a lot of information/data is present here, referring to which in the main text could help build the narrative of the paper. Why not use it in the revised version?
- I believe this paper has lots of potential. There is plenty of interesting information and useful conclusions that it could teach to the reader, but these conclusions are either indirect and buried in the text or not stated explicitly at all. The fact that adding random chemicals to a reaction mixture, repeatedly, leads to a (sometimes less, sometimes more) complex mixture of products is not a revolutionary observation. Complexity for the sake of complexity cannot be the goal here, can it? Alternatively, is the experimental setup the major focus? There are lots of observations made in the text but not nearly enough interpretation. To give an example, on page 14 it is stated “we see that the mass index can increase even when there is not a dramatic increase of the mass of the heaviest product in the cycle” – and this is it. No discussion/interpretation of this observation, even speculatively. Instead, what follows is the Conclusions section, which begins by restating what was done in this study.

This paper requires major revisions. Please articulate directly the following in the text: what

exactly does this study teach us? what is the main point? how can the message be used by prebiotic chemists to improve their approach to experiments?

Taking all the above into consideration, I believe a major revision is required before this manuscript can be published in Nature Communications.

Reviewer #4:

Remarks to the Author:

This communication describes an automated system for running long term chemistry experiments (days to weeks or months). It uses automated delivery of three chemical building blocks / reagents, chosen from a human preselected set of 18 low molecular weight molecules. After a set reaction time 70% of the reaction mixture is removed for analysis and replenished with building blocks to recursively drive the system out of equilibrium on a cycle by cycle basis. A decision algorithm takes MS data from the end of each cycle and compares molecular diversity and higher order molecular weight species using a "mass index" metric to compare one cycle to another. It automatically changes the reagent set delivered according to the rate of change in the mass index. The goal is to over time drive the system to higher molecular weight species as a model of prebiotic chemistry.

As such this is a novel approach to achieve the stated goal of creating a platform for "designing new types of origin of life experiments that allow testable hypotheses for the emergence of life from prebiotic chemistry".

The mass index as a measure of bias toward higher order molecular species is the primary metric used to measure the success of experimental runs. The authors show several long-term runs where the mass index changes over the course of multiple cycles and in some cases demonstrates a clear shift to higher order molecular species. Perhaps not surprisingly given the chemical complexity of the reaction milieu, repeat of identical runs shown in Figure 5 seem to indicate different mass index results over the course of each run, which the authors explain as possibly due to differences in the physical structure of the minerals used in the reactor.

While a laudable approach to start to address this highly complex and burgeoning field in prebiotic chemistry (and perhaps a new arm of chemistry in general), it raises more questions than it provides answers about the kind of data it can produce and how to analyze it. The automated experimental approach is a novel method for handling long range chemistry but is largely missing controls or standards by which to measure and compare data. For example, some kind of control experiment (perhaps through a set of baseline runs, or e.g. like the non-recursive experiment example reported in Doran et. al. *Angew. Chem. Int. Ed.* 2019, 58, 11253 –11256) could help to define just how different the repeat identical runs in Figure 5 are. This may also enable comparison of different run conditions, at least at a systems level. This may be planned for future exploration using the described set up, but some discussion on how the authors might go about this is warranted.

Along these lines, some more detailed discussion on how this platform would be used in future to design experiments around Origins of Life, prebiotic chemistry, or just long range chemistry would help to explain the value of the system.

Comments on writing style: While an interesting paper, the writing is very poor and overly complex at times making this a very difficult concept to get across to readers unfamiliar with prior work. It needs significant editing for English and readability. Secondly the principles of how the system works are difficult to understand without looking at the Supporting Information. It is recommended that a perhaps more detailed version of Figure 19 in the SI be brought into the main manuscript as a way to graphically explain the experimental design and automated decision-making workflow.

Full description of all of the runs shown in Figure 4 is missing. Runs B,C and D would appear to be identical runs if these are the same as described in the SI but this is not referenced or discussed in the text. How do these relate to runs 10,11 and 12 in Figure 5? Notably the mass index for run 10

is the highest observed in this paper, yet not discussed.

The minerals used in the described experiments are not listed in the experimental or SI.

There is some variance in nomenclature for the "mass index" in the main manuscript and "m/z index" in the SI which is potentially confusing.

Reply to reviewer comments. Reviewer comments in italics, our reply in normal type.

Overview.

We thank the reviewers for their very detailed comments and suggestions, and we have examined and addressed every single point without fail. We hope the reviewers are happy with our replies and we are very happy to address further comments and questions.

Reviewer #1 (Remarks to the Author):

The authors present an autonomous robot to run experiments running in several rounds over several weeks in an attempt to study "artificial life experiments". Their platform is a nice setup to study reactions running in multiple cycles. The setup and experimental details are very well documented.

We are glad the reviewer has understood the purpose of our platform and found the experimental details and robotic platform to be clearly explained.

The idea to use it to reproduce prebiotic chemistry is a good one, but the experiments and the conclusions fail to convince me (see "Main concerns" below).

Our goal is not to reproduce the specific chemical reactions that led to the emergence of life on Earth. We understand where the confusion may have come from because this is typically the goal of prebiotic chemical experiments. Instead we are interested in demonstrating that a robotic system can steer chemical mixtures towards higher product complexity. We believe our results demonstrate this and we hope our clarifications below convince the reviewer.

I think that the authors are attempting too much at once, without taking the time to evaluate the single decisions taken for the experiment setup; in particular:

- *the choice of the initial conditions (input molecules and minerals)*
- *the choice of the metrics*
- *the iterative setup of the experiment (dilution problematic below)*
- *the decisions made during the iterative setup*

We appreciate the reviewers concern, and we agree the manuscript contains a lot of different components that need to be explained clearly so that the reader can see how they come together in our results. We have taken opportunity to revise the manuscript to be clearer covering these aspects. We've added a table to the SI to indicate reasons why all of the input compounds were chosen, we have clarified why the particular metric was chosen and how it is related to the iterative aspect of our experimental design.

It is important to note that the choice of the initial conditions, metrics, iterative setup, and decisions are all variable and it is easy to imagine setting up entirely different sets of experiments varying all the aspects outlined above systematically. A key aspect is that we want to ensure others can copy this work which is why the SI outlines how to make the rig, and the control software will be made open and available – so that others who wish to explore a different parameter space can do so.

Also the key thesis presented in the manuscript is to 'explore a prebiotically plausible chemical space' with automation, that can be copied and adapted by others, and we hope that is aspect is one of the key reasons for publication and might convince the referee that urgent publication is needed.

Main concerns:

- *Dilution. The authors explain that at each cycle, 30% of the volume is kept from the previous cycle. This leads to an exponential decrease of the concentration of added compounds in the reaction mixture. Already after 10 cycles, the relative concentration is $0.3^{10} = 6 \cdot 10^{-6}$ compared to when a compound is added.*

The question of dilution is important for the set-up, but again many approaches are possible. The reviewer is right that over time many of the components of the mixture will be diluted to infinitesimal concentrations. The critical exception to this is any compounds which are amplified in the reaction cycle, and this idea is one we were interested in exploring. If the concentration of a particular species is increased during the cycle it will be preferentially retained while all species that do not increase will be diluted out. This is like the CSTR flow set-ups often used in the simulation of prebiotic systems. We have clarified this aspect of our experimental design to say: "By recursively cycling and diluting the product mixture, our experiments were designed to dilute out any compounds not amplified during the reaction cycle, leaving only those compounds which were produced over the dilution, or those that bind to the mineral surfaces to remain"

Considering (from the SI) that on average the added compounds are changed every 10 cycles or so, it means that we can basically consider only the three compounds being added in the current window and in the previous window: everything from previous windows (reactants and products) will be present in negligible concentrations and not likely to be above the threshold for the calculation of the mass index.

This would be correct if no species are being amplified during the reaction cycles. One of the key results is that changing the input reagents automatically did not decrease the Mass Index of the mixture over time, and in fact in many cases adding new reagents increased the Mass Index over consecutive cycles, suggesting the amplification of some species over the net dilution. This is incredibly interesting and requires extensive follow up with precise molecular level mechanistic investigations.

Therefore, if I see it correctly, the experiment is rather about a "moving window" of reactivity, where the observations will mainly relate to reactants being added in neighboring cycles. Considering this, for the observations the total number of cycles is irrelevant:

The total number of cycles are significant for two different reasons. From the robotics perspective, it demonstrates that the platform can perform long term reactions with minimal intervention from the scientist. From the experimental perspective, the reaction cycles give an indication of the length of the chemical history for the reaction. As stated above the reactions retain some compounds in the face of serial dilution, those compounds interact with newly selected reagents and mineral surfaces resulting in the non-trivial behaviour observed over the length of the entire reactions.

even if there are 150 cycles, the observations at the 150th cycle will only "see" things that were added in the last ~10 cycles. To me, this is a major shortcoming and puts a big question mark on the conclusions of this paper, as it prevents any real build-up of molecular complexity.

This is a key thesis as one would expect things to wash out over time. But this is not correct because we show some species are amplified above the dilution or are partially retained on the mineral surfaces. This is because we have seen examples of the Mass Index increasing over the many cycles suggesting that there is in-fact a build-up of chemical complexity. See Figure 4 in the manuscript for example.

- *Mass index. I find this metric very questionable (at least, the paper does not convince me of its usefulness). First, because of the dilution issue mentioned above, one cannot expect the global trend of mass index to be meaningful except in the first ~10-20 cycles, as the initial products will have virtually disappeared.*

As we have stated above the serial dilution via recursive cycles will only remove those species that are not amplified in the reaction cycles. The Mass Index was designed with this in mind. The metric is higher when the largest mass observed is higher and when the number of other observed products is lower (meaning

more species are diluted out). Taken together this gives us a clear way to identify mixtures that are less diverse (or messy) but which still have complex products.

Second, the paper does not convince me that this metric is actually a good descriptor of what is happening in the experiment. I do understand the need for simple observables when observing mixtures, but it is difficult to take this metric seriously without explicit demonstration or proof of the usefulness based on an actual analysis of the products.

The work of Doran *et al.* 2019 (Angew Chem Int Ed Engl 58, 11253-11256, doi:10.1002/anie.201902287) showed that this metric can be a powerful tool in measuring the change in a complex product mixture. In addition, we have provided examples of how the metric works can be found in the SI section 3 as well as a list of identified products from the sample mixture which can be found in section 4.2 of the SI. We are glad the reviewer agrees that simple observables are required for mixtures and we think developing a more advanced set of metrics is an exciting direction of future research that we are working on.

Without this, I can only look at the formula, and notice that it is, in my opinion, not very robust: a little change in the MS intensity (either from one cycle to the next, or because of experimental noise) can lead to one species being above the threshold or not.

Thresholding mass spectra is a common technique used routinely in almost all analysis. It is essential to remove electrical and experimental noise. While it could cause changes in the Mass Index, thresholding can cause changes in any measure based on mass spectrometry. The goal with our metric in this work was to provide a simple measure to make automated decisions about future reaction cycles. We believe the Mass Index accomplished this within the scope of our experimental design and details can be found in section 5.4 of the SI. We have done extensive control experiments to establish the robustness of the approach, and despite the fact the system is sensitive to noise, see Figure 26 in the SI, we were able to verify that the experimental procedures were still robust by showing that many product ions are reproduced in experimental repeats. Using this automation approach, and building on the use of an automatic assay is vital to make progress to determine a more robust scale to measure complexity of compounds and mixtures to help us achieve our long goal of using long term experiments such as this to search chemical space for de-novo artificial life forms.

Such a little change in intensity can therefore lead to a big jump in the value of the mass index because the mass range or the number of considered peaks will suddenly be different. This may be one of the reasons why the reproducibility test failed.

The attempt at reproducibility did not fail, as the data in figure 5 shows. The variation came in the Mass Index calculation, which is a sensitive system. Small ionization differences can make a difference in the metric especially, with a mass spectrometer that runs 24/7. But every precaution was taken with an above described thresholding and a regular cleaning and calibration procedure.

- High initial values of m/z. The article is about the build-up of molecular complexity. There, anyone would expect the highest m/z value as a function of the cycle to be increasing ("looking for an increase in the mass of the product species").

Our hypothesis was that the complexity of the products would increase over the course of experimental cycles. In this work we tested this idea, and increasing complexity was observed in Run A (manuscript Figure 4) but turned out not to be the rule. This is because of experimental design where the input composition was randomized – not all combinations can lead to an increase. The Mass Index not only takes the increase in mass into account, but also the count of the product species, as this is an equally important value evaluating the product mixture.

The initially very high values of m/z, however, prevent any subsequent conclusion: how can one believe any of the following m/z values (or mass index values) if the initial solution is contaminated with complex molecules already?

The initially high m/z values in the first five cycles are thought to be due to materials being washed out of the minerals, despite the pre-washing procedures in place. Mineral leaching can be seen in the ICP data which was added to the SI section 2.2. These compounds were not amplified through the recursive cycle and therefore not retained through the entire experiment.

This makes me question all the subsequent results (including the mass index). The authors guess that this may be related to the added minerals. In this case, why not try the same experiment without those minerals?

We thank the reviewer for this suggestion, and we have performed the additional controls and added them to the SI. In addition to the ICP data we have added data from a mineral control of run G to the SI in section 4.1.3. In that control we compared runs with and without minerals, as well as analysis of water stirred with minerals under the same conditions. The base peak chromatograms of these runs showed some peaks, but these peaks were consistent throughout 168 hours of stirring and heating. This shows that the Mass Index would not be changed throughout a prolonged experiment by the use of minerals, excluding the initial wash out reported.

- Number of product species. I am struggling to understand how there can be more than 20000 product species starting from only three compounds, and how one can give any meaning to this value.

This number of product species was caused by very cautious thresholding. This was now further improved to pick up less noise. The used thresholding is detailed explained in the SI section 5.4.

- Combined with the previous concerns, I find the lack of any actual analysis of the content of the reaction mixtures, in terms of chemical structures, very disconcerting. This would actually be a very interesting piece of information! Especially if one sets out to reproduce reactions leading to life, anyone would be curious to see what is happening, even if only few examples were given. a

We have attempted to manual ID the reaction mixtures of one exemplary run. The chemical structures have been determined based on the experimentally found massed and are listed in the SI section 4.2.

The abstract even says "explore unconstrained multicomponent reactions" - I do not consider the analysis of a single metric to be an actual exploration. Only measuring an increase in complexity is not as exciting and not as unexpected.

Our platform allows us to explore the dynamics of these unconstrained reaction mixtures in ways that were not previously possible. The specific algorithmic implementation used here was simple, by design, but our platform allows us to store samples for off-line analysis as well. We have done that in several cases as for example with the analysis of offline samples on the Orbitrap which produced data that was included in figure 5 in the main manuscript as well as data analysis on the Bruker Maxis which is described in detail in the SI section 4.1.6. We also performed a comparison of other algorithms on already collected datasets from the Advion described in section 5.4 of the SI. These analyses constitute an exploration of the mixtures.

- I would also find some basic checks in terms of reactions and chemical structures reassuring, given all the points above: it could at least show that the system is working as expected, and for instance give further insight about the large initial m/z values or the peaks at 240, which are not explained. It would also show that the authors actually understand the chemistry that is happening.

To help provide insight into the chemical identities of the products, we made tentative formula assignments for features in the Bruker MS data (see SI section 4.2.1). We also compared these to the theoretical products predicted from the molecular transformer. Of the 429 product molecules calculated, 22 were confirmed to

match with proposed formulae from the MS data (see SI section 4.2.2). As we stated above, the large initial m/z values were likely due to mineral wash out, which was analysed by ICP and can be found in section 2.2 of the SI. This data shows that the raised m/z values in the beginning of the experiment around cycle 1-5 correlate with the leaching of elements of the mineral ulexite. We hope that these further analyses and additions to the SI are enough to convince the reviewer that there is an understanding of the chemistry, but we reiterate that the approach taken and the conclusions of this work do not require the identification of the products.

- Reproducibility. The failed attempt at reproducibility is worrying. It was even set as one of the goals, "to do this in a reproducible way". But the data shows that this is not the case: the curve with the mass index is significantly different!

As mentioned above, the attempt at reproducibility did not fail as the data in figure 5 shows. The linear correlation in these graphs prove that the product mixtures produced with the automated platform are of comparable composition therefore showing that the experiment was successfully reproduced. The variation came in the Mass Index calculation, which is a more sensitive system. Small ionization differences can make a difference in the calculation and with a mass spectrometer that runs 24/7, even with regular calibration, uniform performance is a challenge. Defining metrics that are more robust is an interesting question for future work. It is important to present this data so others who intend to follow the work will be able to honestly work from our foundations.

Other general comments:

- I am not sure what to think about the conclusion on the ability of the system to do "long term reactions" (in the title!). Because of the dilution issue mentioned above, I am not sure that one can consider this to be correct.

As we have discussed above, the experiments retain some chemical history of the larger number of reaction cycles. Further even in the case that most of the chemical species are diluted out, the number of consecutive cycles provides an indication of the robustness of robotic platform.

At most, it shows that the software and hardware integration is robust. Also, when reading the title, I was expecting "long" to mean several months or years.

Showing that the software and hardware integration is robust is an important part of the project. Further in comparison to the majority of the experiments published, the presented runs can be titled "long term". With the platform presented longer experiments will be feasible.

- How were the 18 starting mixtures chosen? The authors write a full paragraph that sounds a bit defensive about them being chosen by humans, but I would be very interested to know why these 18 specific compounds were chosen.

The starting material library was mostly chosen by purposefully avoiding specific chemical groups, meaning no "common" autocatalytic cycle precursors, sugars or amino acids have been selected. The aim was to solely concentrate on small functional building block molecules with no immediate connection or function. We added a table with each starting material and their properties to the SI section 2.1 to further specify our decision.

- I find that the paper over-emphasizes the "origin of life" aspect of the work. The choice of the 18 starting mixtures is not justified, and no single product molecule is given.

We have argued that understanding the origin of life, and attempts to assemble *de novo* life, will require novel experimental platforms, like the one developed here. This is because the most prominent theoretical ideas and computational models make assumptions about the dynamics of unconstrained multicomponent

reactions over long time scales. Without experimental platforms like the one we have demonstrated these assumptions cannot be experimentally tested. Rather than focusing on particular biomolecules, we have demonstrated that a multicomponent mixture can be steered through a recursive cycle.

- In the abstract: "We show that the robot can discover the production of high complexity molecules": there is no proof of any high complexity molecules; the paper does not contain a single chemical structure except the 18 starting compounds.

Figure 1 shows chemical structures of products proposed by the model. The figure was now improved for better readability. A list of proposed formulae for product molecules of run G which have been calculated based on the measured m/z value has been added to the SI section 4.2.

- In the abstract: "experiments that allow testable hypotheses for the emergence of life from prebiotic chemistry": I don't see any such hypothesis in the paper, and it is not explained how they would be testable, even if the system allowed for reproducible experiments.

While this approach is an important step in designing new types of Origin of Life experiments that allow testable hypotheses for the emergence of life from prebiotic chemistry, we agree that the above sentence was probably not appropriate and have removed it.

- "we first explored the chemical space accessible to the 18 input reagents computationally". Such a good idea, but what a pity that the authors do nothing more with it than seeing that there are 2206 possible reactions.

We are glad that the reviewer thought it is a good idea to explore the chemical space computationally. To do more with the gained insight into the chemical space, we compared the 429 computationally found product species (multiple reactions led to the same product) with experimentally found product formulas and have been able to find 22 matching compounds. These compounds are listed in the SI section 4.2.2.

- At the end of each cycle, 70% of the solution is stored. The experiments consist in 60-150 cycles, but on the picture I see only ~20 vials for storing, which means that either they were changed by hand, or not all the cycles were then stored.

This observation is correct, up to 20 vials have been stored automatically until they were changed by hand. We have further specified this in the manuscript.

- "not only automate a vast number of experiments, but also make decisions on the fly about which routes to follow". It would be great to inspect other ways to make decisions than to rely only on the mass index - and I find the decision based on the slope of the mass index somewhat arbitrary.

Other algorithms to assign the data have been tested and a detailed description and comparison of these has been added to the SI section 5.4.

- "The data shows that the algorithm succeeds in controlling the experiment, leading to different behaviors of each experiment while not generating the chemical mess, the simulation showed." The authors do not explain how the chemical mess is avoided - this is not visible in the data.

By chemical mess we mean unconstrained combinatorial explosion. The Mass Index metric falls to lower values in the case of a combinatorial explosion, so the fact that the algorithm successfully steers the experiment towards rising Mass Index, shows that this state has been avoided, see: Doran *et al.*, Emergence of Function and Selection from Recursively Programmed Polymerisation Reactions in Mineral Environments. *Angewandte Chemie International Edition* **58**, 11253-11256 (2019).

Also, if the authors base this statement on the computational approach, they should take in consideration that in their experiments, only a subset of the product molecules will be possible at any time (dilution issue

above), and that several products may have the same chemical formula and therefore lead to a single peak in the MS spectrum.

As stated above, we are basing the statement on the behaviour of the Mass Index metric and the observed rise over the experiment. Regarding the possibility of single peaks masking multiple compounds, the methods used include liquid chromatography and so the features are based on both retention time and mass, much reducing the possibility of this problem.

Details:

- in the abstract: the size of the data (20 GB) is irrelevant.

We don't think this is irrelevant since data store will be important for others taking on this approach, however this has been amended in the revised manuscript.

- Why did the authors do the reactions under a nitrogen atmosphere? Was it a deliberate choice?

Yes, this was a deliberate choice to avoid potential oxidation of the products and have a clear control over the experimental atmosphere. A note of this has been added to the manuscript.

- "just 3.7% of all experiments reported to Reaxys": considering the size of Reaxys, this is a lot of reactions! I wouldn't take this as an argument that there are few long reactions reported in chemistry.

We agree that this does not mean there are few long-term reactions reported in Reaxys. The aim was rather to point out that this field might be underrepresented than other chemical reaction types.

- Figure 1 is would be more readable if it was larger.

Figure 1 was increased in size and the readability should now be improved.

- Page 7: "known to be too complex to be analyzed". I would expect further clarification at this point.

The mixtures generated by the platform are very complex. It would be intractable to analyze each sample using the techniques typically employed by synthetic chemists. We've clarified this in the manuscript to say "which is a chemical system known to be too complex to be analyzed conventionally^{6,26,27}, with identification of all the product species involved. We address the difficulty of analyzing each specific product in these mixtures by taking the alternative 'systems' approach of observing the behavior of the chemical mixtures and looking for global phenomena in the recorded data, rather than concentrating on targeted analysis."

- "Taking in account how the index is calculated and that especially heavy masses are weighted higher, the data is complementing itself" What does this mean?

This was meant in the context of the reproduced runs. Figure 5 shows that with increasing m/z values, the Mass Index is different for each run, as the Mass Index is more sensitive to be changed by heavy masses.

- "It could be argued that the algorithm is too simple and cutting too much of the actual data, but when testing alternative algorithms, we observed that all of them have been complementing each other" What other algorithms? What do you mean by "complementing each other"?

This has been clarified in the manuscript to say "It could be argued that the algorithm is too simple and cuts out too much of the actual data. To investigate this, other algorithms looking into information entropy or the weight by intensity values have been tested (detailed description of each algorithm can be found in the SI). We observed that the algorithms complemented each other, as for example in run E (Figure 4) cycle 120, the Mass Index algorithm and the mass by intensity algorithm values increased while the information entropy value dropped. This indicates a drop in unique species while the m/z values increased. While this showed that the algorithms worked in general, we have not been able to get further insights into the actual data by using different algorithms."

- *"as well to enable a calculation in a short amount of time": this is not a good argument. Computers can analyze huge amounts of data in seconds!! This is nothing compared to the weeks of the experiment.*

It is true that computers can analyse huge amounts of data, but this statement is made in context of MS data. The processing of large LC-MS files can still be a challenge for computers today, particularly when the samples are complex. Typical computation times can vary from seconds to hours if the sample is complex enough. Our method was so simple that it was guaranteed to be computed in an order of magnitude less time than cycle time. In future experiments we would like to perform more interesting analysis using the online analytical tools.

Typos and languages:

- *everywhere: please review the language. I understand that the first author is not a native speaker and I would recommend a native speaker to do extensive proof-reading. Many formulations are strange. Also, a lot of commas are placed where not necessary in English (f.i. page 12 "There are several reasons, why differences").*

We thank the referee for pointing out the issues with the language and we have checked and modified the language as needed.

- p. 3: *"now only" -> "not only"*

This has now been corrected in the revised manuscript.

- p. 3: *"a heterogenous reaction environments" -> remove "a"*

This has now been corrected in the revised manuscript.

- p. 5: *"4 syringe" -> "4 syringes"*

This has now been corrected in the revised manuscript.

- p. 5: *"purpose- build" -> "purpose-built"?*

This has now been corrected in the revised manuscript.

- p. 6: *"Everything was stirred and heated until the stirring was terminated one hour before the sampling was due" this is not very understandable*

This has now been corrected in the revised manuscript.

- p. 5: *"which choose" -> "which chose" / "which chooses"*

This has now been corrected in the revised manuscript.

- p. 8: *"bespoke" looks strange in this context*

This has now been corrected in the revised manuscript.

- p. 8: *"By calculating this number, we are able to use it as a metric, which enables the comparison of cycles with each other, fully automated": please reformulate*

This has now been corrected in the revised manuscript.

- p. 8: *"randomized" -> "randomly"*

This has now been corrected in the revised manuscript.

- p. 8: *"As well as the index can" -> "Furthermore, the index can"*

This has now been corrected in the revised manuscript.

- p. 8: *"Experimental data, of the mass index, their slope" what does this mean?*

This has now been corrected in the revised manuscript.

- p. 8: *What is a "hill climber rise"?*

We have clarified this in the manuscript to say that "four of the six experiments show periods where the Mass Index progressively increases, even as the selected inputs are changed. This means that a heavier product was forming, and/or the number of overall peaks over threshold was declining, while the heaviest mass remained."

- p. 10: *"We wonder" -> "We wondered"*

This has now been corrected in the revised manuscript.

- p. 11: *"We can see that these experiments...": long sentence that is not very readable*

This has now been corrected in the revised manuscript.

- p. 15: *"The reactivity of the product library was simulated, and the reagent network showed, the reagent library was prone to create a messy product mixture, combined under the used conditions" not very understandable*

This has been rewritten to make the point more clear.

- p. 16: *"HPLC-grad" -> "HPLC-grade"*

This has now been corrected in the revised manuscript.

Reviewer #2 (Remarks to the Author):

Cronin and co-workers describe an automated laboratory system for long-term experiments setup to explore unconstrained multicomponent reactions. Their system's acquisition of data from mass spectrometric, ability to adjust reagent input based on such data and user-defined algorithms allow these researchers to explore the immense chemical space of products that is possible with the input of eighteen, and possibly more, organic compounds and minerals. Results are provided from runs that were carried out for up to 4 weeks with 150 algorithmic decision cycles. These experiments provide validation that their automated system, as intended, is able to generate and deal with the extremely large data sets that result from recursive-unconstrained chemical reactions.

We are glad the reviewer has understood the purpose of our platform and agrees that the experiment validates the platform and its ability to process the analytical data.

Overall impression:

The authors state that running long-term, and potentially open-ended, experiments is an approach to origins of life, prebiotic chemistry, and artificial life research that is necessary but currently lacking from these fields. I agree.

We are very happy that the reviewer agrees with this approach being necessary in the field.

These fields need such approaches that break from the more standard approaches of synthetic organic chemists and prebiotic chemists who have traditionally sought to produce individual compounds through carefully designed and controlled reactions. In contrast, the approach being advanced by the Cronin and coworkers minimizes human bias to the selection of reagents and reaction conditions.

We absolutely agree with the reviewer and hope that our approach will eventually be embraced by the wider community.

After that, the chemistry of the system is allowed to follow its own path, with some direction provided by a relatively simple metric, such as m/z distribution.

Exactly, this is precisely what we have set out to do. The automation and feedback then allows the trajectory of the system to be followed and steered.

While I appreciate the motivation behind the work described in this manuscript and the potential for the automated system developed to reveal chemical transformation and synthetic routes that might otherwise never be revealed, actual chemical results presented in this manuscript remain rather obscure.

Our results demonstrate that an unconstrained chemical mixture can be steered algorithmically using a robotic platform. While we imagine a platform of this type could help identify novel synthetic pathways, it is not our goal here. Instead we are interested in demonstrating that we can use the platform we developed to perform long term experiments to explore the dynamics of these unconstrained mixtures. Our results show that using a set of simple precursors we can steer the system towards more complex states.

The data provided show that recursive experiments guided by autonomous computer control can lead to long term increases in mass index, for example.

Yes, this is interesting because it shows that the mixture can make more complex materials from simple inputs, and that such mixtures can be steered by the autonomous system.

However, without a revelation and discussion of the actual molecules that are produced from these processes it is not possible to appreciate how interesting or unexpected the chemistry is that emerges from this non-rational approach.

We have added more chemical analysis to the manuscript, including new non-mineral controls in SI section 4.1.3, and ICP analysis of the initial product mixtures to SI section 2.2. Further we have identified some of product formulas and highlighted that 29 of the observed formula were predicted to exist from the model of the chemical reaction network. With these additions we believe we have added more chemically relevant details.

In this context, I feel that this manuscript is better suited for publication in Nature Methods or Review of Scientific Instrumentation.

Our results are interdisciplinary in nature, drawing on robotics and analytical chemistry and they are of interest to the origin of life community (which is a diverse audience itself). We think our results are of interest to the broad readership of Nature Communications.

Specific issues to address:

The word "prebiotic" should be removed from the title, are at least qualified as a potential application of the instrument profiled. The authors note several times that it is not their intention to show prebiotic chemistry in this work, e.g. in the conclusion: "The automated system was tested using a simple heuristic analysis of complex mixtures based on a system level prospective, rather than focusing on a narrow set of "prebiotic plausible," or "biologically relevant substrates". Also on page 6 "We tried to concentrate on the function of the building blocks itself instead of taking prebiotic chemistry narrative in account (No "common" autocatalytic cycle precursors, sugars or amino acids)." Similar statements are made in the SI. The work may have potential future applications relating to abiotic chemistry, but no such arguments or data are presented in this work.

The work presented is all about searching for complexity arising from simple building blocks through cycles. While this could certainly be seen as part of a prebiotic / Origin of Life scenario, that's not the focus. However, when we use the term 'prebiotic' in the title, it is as part of 'robotic prebiotic chemist' and within this work we aim not only to demonstrate our autonomous platform, but also to challenge what it is that a prebiotic chemist does. We accept that the tool we have built does not quite fit the prevailing prebiotic narrative but suggest that these kinds of 'messy' experiments with systems level analysis is where the future of prebiotic chemistry is headed.

There are several sentence fragments, typographical errors and use of colloquial language throughout the body of the text. As two examples, the first line of the Results reads "In this work we set out to design a system that can now [not?] only automate" and on page 8, it is stated "experiments presented follow some point a hill climber rise"

These have been corrected in the revised manuscript which we now think has been improved.

Strictly speaking, the term "mineral" refers to a natural substance. Is that the case for all of the inorganic salts that are referred to as minerals. If not the point should be made that the minerals used are synthetic. Otherwise further confirmation of mineral purity may be necessary to guard against natural substance heterogeneity contributing to experimental variations.

The reviewer is correct in pointing out that the term mineral should be strictly used solely for natural substances and a clear distinction must be made between minerals and inorganic salts. In this work, solely natural minerals have been used and the source as well as the wash procedure is described in the method section of the manuscript. Homogenization of bulk mineral samples before the experiment, by crushing / grinding and sieving, ensures that sample heterogeneity is unlikely to cause issues. To further prove that mineral impurities did not change the outcome of the experiments, a mineral control experiment was added to the SI in section 4.1.3. In this mineral control experiment, we compared runs with and without minerals, as well as analysed water stirred with minerals. The base peak chromatograms of these runs showed some peaks, but they were consistent throughout 168 hours of stirring and heating.

Is the pH of the reactions measured or controlled? This information could be important, as it may alter the kinetics of potential reactions, and possibly the mineral solubilities (altering the effects of potential "surface" chemistry).

The pH was neither measured nor controlled in the initial experiments. We had stored samples offline and we have now tested the pH of stored samples of run G, which can be found in the SI. While comparing the pH values with the Mass Index, no correlation was visible but we agree with the referee that this information could be useful and an implementation to measure the pH in future experiments would be interesting.

Were the minerals themselves replenished after each cycle? What controls were there for this? It seems like continual addition would lead to a build-up of solid material in the reaction vessel.

The minerals were only added to the initial cycle and kept on the bottom of the reactor throughout the experiment.

The authors state that further analysis on the effects of minerals could be the subject of future work, but more information about how the mineral content was controlled throughout this study is still needed. Further information about the minerals and the amounts used have been added to the methods section of the revised manuscript.

The authors state that the cycle times vary from "3 to 12 hours", but no information is given about the duration of cycles during any individual runs. This duration information is important for reproducibility of the results. For example, on page 10, all of the input changes are laid out for a given experiment (summarized in

Figure 5) that was repeated 3 times, but information on the cycle time is not provided. I would assume that cycle times are the same, but these need to be stated explicitly.

This has now been further clarified and the cycle times of each mentioned experiment has been stated explicitly. In the reproducibility experiment the cycle time of 6 hours has been indeed the same in all 3 runs.

Also, the times required for HPLC-MS runs could be important if this analysis must be completed before the next cycle/ input change is allowed to proceed.

This is correct, the HPLC-MS time is the most important factor when making a decision between cycles. In the methods section we detail that the LC time was 40 minutes in total, with 20 minutes analysis and 20 minutes column cleaning.

For Figure 4 what is the significance, if any, of the spherical data points in the 3D plot?

These points show the relationship between the Mass Index against the number of species per cycle. This has been clarified in the manuscript.

Figure 5: The figure caption labels refer to the incorrect positions in the image (right/left etc.)
We thank the reviewer for noticing this error. It has now been corrected in the revised manuscript.

Figure 5: Were runs 10, 11 and 12 identical replicates?

Yes 10, 11 and 12 are identical replicates, meaning that run 10 was allowed to proceed under autonomous control as described while run 11 and 12 did not make automatic decisions but simply followed the decisions previously determined during run 10. This has been further highlighted in the revised manuscript.

Figure 5: Are the linear plots suppose to show that there is a correlation perfect or is a straight line fully expected due to the nature of the data being plotted? If the latter, different presentations of this data should be used so as to emphasize what is interesting in the data being compared.

We thank the reviewer to point out that this representation of the data was not particular useful to show the different features of these runs. The straight line was on purpose as the difference was in the disruption of the lines, but as this did not highlight the overlap and differences between the runs. The figure has been changed in the new version. We hope the changed figure makes the representation of the differences and common features of the reproduced runs clearer.

Page 12: As a possible explanation for the different result obtained for replicates of the same experiment, the authors state: "In addition to this, each MS run, even of the identical sample, will change slightly through ionization and can lead to a slight variation in the overall detected product species of the run." Can a reference be provided for data that supports this claim (perhaps using standards). It would seem that if the solutions had identical concentrations of compounds, their ionizations would not differ. Also, variances in mineral composition are mentioned as a possible reason for non-reproducibility. This argument could be likewise be fortified with references or control experiments.

In our experimental setup, the LC-MS technique needed to host a range of different matrixes as cycles of one experiment could have varied widely in terms of the acid / base environment, more / less inorganic salt present from the input composition or more / less adhesion to minerals depending what molecules were present in the product mixture of that cycle. Usually, the instrument would be tuned for a specific matrix, but the nature of our experiment makes that impossible and with so many variables, a variation in ionisation of these different product mixtures can occur even if the concentration is consistent. We suggest the following reference as an example of this: <https://doi.org/10.1016/j.clinbiochem.2004.11.007>

In addition, we have analyzed some samples offline, using higher resolution instruments, which is shown in figure 5. This data shows that some new peaks are present in the samples but there is a large number of peaks present across the experimental repeats.

Page 13: The authors state "It could be argued that the algorithm is too simple and cutting too much of the actual data, but when testing alternative algorithms, we observed that all of them have been complementing each other, but have not been able to get further insides about the actual data" What other algorithms were tested?

A detailed description and comparison of each of the used algorithms has been added to the SI section 5.4 as well as the statement in the manuscript was further expanded.

In the Conclusions the authors state "The data shows that the algorithm succeeds in controlling the experiment, leading to different behaviors of each experiment while not generating the chemical mess" Do the authors mean to say "the chemical mess is reduced when compared to the total number of possibilities"?

By chemical mess we mean unconstrained combinatorial explosion of products. The Mass Index metric (see reference Doran *et al.*, Emergence of Function and Selection from Recursively Programmed Polymerisation Reactions in Mineral Environments. *Angewandte Chemie International Edition* **58**, 11253-11256 (2019)) falls to lower values in the case of a combinatorial explosion, so the fact that the algorithm successfully steers the experiment towards rising Mass Index, shows that this state has been avoided. We have clarified this in the manuscript.

Could this also be partially due to the 10^6 cut-off imposed during mass index calculation? If so, this should be acknowledged.

Thresholding mass spectra is a common technique used routinely in essentially all analysis. It is essential to remove electrical and experimental noise. While it could cause changes in the Mass Index, thresholding can cause changes in any measure based on mass spectrometry. This cut-off was carefully chosen to only exclude instrument noise and examples of this are shown in section 5.4 of the SI.

Page 19: pressure units/or gas flow rate units missing

This has now been corrected in the revised manuscript.

SI Figure S5 and S6: Sulfate anion (?) structure is incorrect.

The structure has now been corrected in the revised SI.

SI Figure S8: There are two Resorcinol chromatograms. Is one mislabeled? Additionally, the top Resorcinol chromatogram look identical to the pyruvic acid chromatogram.

This was indeed a mistake and has now been corrected in the revised SI.

Reviewer #3 (Remarks to the Author):

Gaining insight into navigating complex chemical networks and searching for a functional increase of complexity is of utmost importance in modern prebiotic chemistry which takes a step beyond the classical synthetic organic chemistry approach based on individual reactions.

Yes, we are happy that the reviewer agrees with us.

This paper by Asche et al presents an automated platform that allows for multicomponent reactions to be run over long periods of time and in a large number of substrate addition-analysis cycles. The outcome of each cycle is monitored by HPLC-MS, and the obtained MS data are fed to an algorithm which can randomly alter the substrate input based on how much the mass of the product species increases.

The premise of the paper is extremely interesting, especially in light of the gradually shifting paradigm in prebiotic chemistry, which now begins to notice the importance of reaction networks and systems chemistry over the classical stepwise syntheses.

We thank the reviewer for the assessment of our manuscript and are glad that they find the premise interesting.

However, despite the very interesting context, this paper leaves me somewhat confused. One thing I am missing here is an explicit statement why this study was undertaken. Even if it was something along the lines of “we wanted to mix random chemicals to see what happens” (which I do not believe was the case), this should be stated already in the abstract (i.e. what is the existing problem and why it needs solving). Otherwise, as a first-time reader, I have no idea what the authors are trying to tell me and what the implications are. Even the authors’ own previous work is not directly used to frame the discussion (it is hidden in the references)! Secondly, stating the working hypothesis is just as important because without it the manuscript is a collection of data, without a clear storyline.

Our hypothesis was that the complexity of the products would increase over the course of experimental cycles. In this work we tested this idea and increasing complexity can be observed in Run A in Figure 4, but it turned out not to be the rule. With this manuscript we demonstrate a new way to do prebiotic chemistry. By focusing on the algorithmic aspects of the experiment we remove the implicit bias about what important molecules or reactions are relevant to the emergence of living systems. Instead we showed one way to explore the emergence of complex molecules from simple precursors. We have clarified the manuscript to reflect this.

As if referring to the chemistry of the experiments presented here, the language of the manuscript is rather messy in places. Often, the manuscript contains compound sentences with multiple levels of clauses. This leads to two problems. One is that sometimes a preposition or a verb might be missing here and there, and the sentence sounds quite bizarre. Another is that much of the paper’s message becomes lost—and this problem becomes particularly frustrating in the abstract, which is the weakest part of this paper writing-wise. Still, I believe this can be easily fixed.

The manuscript has been extensively revised, and we hope the referee can see that the manuscript has been improved.

Another problem is that the paper still requires some work in terms of phrasing the story and explaining the premise of the study in a way accessible to a general chemist. Many details are missing, some important information is only mentioned briefly (e.g. the choice of reagents/catalysts), but the context/rationale behind it is left to the imagination, or explained in a different section (with no reference to that section).

We have taken care to refine the explanation of our motivation and hypothesis in the revised manuscript. The choice of reagents has also been further explained in the manuscript and in section 2.1 of the SI.

If certain data or details or procedures are presented in the SI, references to appropriate Tables/Figures are needed in the main text.

We have added references to specific sections of the SI to help make important information clear

Finally, sometimes a claim is made but not supported with a reference to the literature. I will list all these issues below

Specific comments and suggested corrections – in a chronological order:

- Page 2 (top): “massively parallel experiments” – I am not sure what this means. A large number of experiments performed in parallel?

Yes, this has now been corrected in the revised manuscript.

- Page 2: *“life was thought to have emerged ca. 3.8 B years ago” – a reference is needed*

A reference has been added to the revised manuscript.

- Page 2: *“Previous approaches to prebiotic chemistry, which have their origin in synthetic organic chemistry, intentionally try to limit the accessible size of the chemical space in experiments” – while in principle I agree with the authors’ sentiment here, limiting the size of the chemical space does not always have to be a bad thing. If the focus of a study is to investigate a mechanism or a certain reaction on a certain functionality, it only makes practical sense to limit the number of variables (or—to “minimise the noise”). Whether the said reaction/mechanism is texted in a broader context afterwards is, of course, another story, and I agree the synthetic organic chemistry school in prebiotic chemistry do generally refrain from this.*

We agree, and as the reviewer says, it is the framing of the carefully orchestrated transformations in the wider context that is often lacking. This is partly due to the idea that what outcome / end product is of interest is known in advance. We try to take a broader approach and accept that we do not know what outcomes are more interesting than others in terms of chemical structure, but that increasing complexity and information content are good targets.

- Page 2 (bottom) – *In this context I suggest also citing the following:*

- *Vincent et al. Chemical Ecosystem Selection on Mineral Surfaces Reveals Long-Term Dynamics Consistent with the Spontaneous Emergence of Mutual Catalysis. Life 9, 80 (2019).*

The reference has been added to the revised manuscript.

- *Baum, D. A. The origin and early evolution of life in chemical composition space. J Theor Biol 456, 295–304 (2018).*

The reference has been added to the revised manuscript.

- *also current ref. 4 seems suitable to be mentioned here as well.*

The reference has been added to this statement in the revised manuscript.

- Page 3 – *“a system that can now only” – should be “not only”*

This has now been corrected in the revised manuscript.

- Page 3 & 4 (Chemical space) – *an explanation is needed why these molecules were chosen and not others, and what a “possible reaction” means in this study. Thermodynamically possible? The rationale behind the choice of molecules is indeed hinted at on page 6/7 in the “Data from the platform” section, but in my view this is far too late. I suggest describing the choice of reagents early in the “Chemical space” section.*

The starting material library was mostly chosen by purposefully avoiding specific chemical groups, meaning no “common” autocatalytic cycle precursors, sugars or amino acids have been selected. The aim was to solely concentrate on small functional building block molecules with no immediate connection or function. We added a table with each starting material and their properties to the SI section 2.1 to further specify our decision. The description of the choice of reagents has been moved to the “Chemical space” section in the revised manuscript.

- Page 4 (top) – *“any combination that gave a score greater than 0.8” – to make this paper accessible to a general chemist, a brief explanation of how this works and what this value means is due here. Otherwise this number is meaningless.*

We've updated the main text to read "Briefly, the molecular transformer is a machine learning model that takes input reagents as arguments and suggest possible product species, the model has a built in mechanism to assess (or score) the quality of its prediction on a scale of 0.0 to 1.0, with reactions scored as 1.0 being the most supported by observed reactions."

- Page 4 – Figure 1 – it is impossible to see what the molecules are even at 400% zoom. Consider resizing.

We thank the reviewer for noticing this formatting error. Figure 1 has now been resized in the revised manuscript.

- Page 5/6 – heterogeneous mixtures and minerals are mentioned, but there is no reference to any table/list of these heterogeneous substances/minerals. How were they chosen?

The minerals and their amounts used has now been added to the methods section of the revised manuscript.

- Page 6 – the sampling is limited to whatever is dissolved in the aqueous solution, but is there any indication of products adsorbed on the mineral surface? It would be a good idea to subject the minerals to surface analysis. Analysis by MALDI-TOF-MS is suggested as future work on page 14, but based on how many times the minerals are mentioned in the paper, at least a qualitative XPS/XRD would already be very informative.

We agree that this would be very interesting to investigate in future, but this is not focus of this study as we concentrated on analysing the change of the product mixture by LC-MS.

- Page 6/7 – "We tried to concentrate on the function of the building blocks" – what function exactly? How/why was this function selected? Could this be elaborated on?

This has been further discussed and the properties of the input materials have been listed in section 2.1 of the SI.

- Page 7 – "a chemical system known to be too complex to be analyzed" – a further comment is needed or at least some references to back up what is "known". Perhaps these (and refs therein)?

- Schmitt-Kopplin, P. et al. Systems chemical analytics: introduction to the challenges of chemical complexity analysis. *Faraday Discuss* 218, 9–28 (2019).

- Geisberger, T. et al. Evolutionary Steps in the Analytics of Primordial Metabolic Evolution. *Life* 9, 50 (2019).

The mixtures generated by the platform are very complex. It would be intractable to analyse each sample using the techniques typically employed by synthetic chemists. We've clarified this in the manuscript to say "which is a chemical system known to be too complex to be analysed conventionally^{6,26,27}, with identification of all the product species involved. We address the difficulty of analysing each specific product in these mixtures by taking the alternative 'systems' approach of observing the behaviour of the chemical mixtures and looking for global phenomena in the recorded data, rather than concentrating on targeted analysis." We thank the reviewer for suggesting these references and added them as well as others to the section in the revised manuscript.

- Page 8 – the meaning of different Mass Index values with respect to the increasing/decreasing product complexity and the number of product species is described in a rather confusing way. Presenting general rules in a tabularised form might be helpful.

We have clarified the grammar in the section describing the Mass Index and the way it trends with larger/small product species and with the total number of product species. In addition we have added this table to the SI section 3.2 to help readers understand the different implications of the Mass Index rising or falling.

	Largest Product Constant	Number Product Species Constant
Mass Index UP	Fewer Product Species	Largest Product Increased
Mass Index DOWN	More Product Species	Largest Product decreased

Commas in random places sometimes change the meaning of sentences—please check this.

The revised manuscript has been proof-read extensively and is now much improved.

- Page 9 – The Mass Index values, as well as the number of product species in each run could be added to Figure 4. Presenting the graphs in 2 columns and 3 rows, with these values added, instead of the current 3 columns x 2 rows, would make Figure 4 much more accessible to the reader.

We changed the layout of the figure to improve readability in the revised manuscript. Taking in account that there are 60 Mass Index and product species values for each plot, adding the numbers would make the figure impossible to read.

- Page 10 – how I understand it, the number of cycles the system is subjected to after the addition of the starting materials is not a set value, and depends on whether the slope falls below the threshold or not. Is this correct? It is not immediately obvious from the way it is described.

That is correct, there is a minimum number of cycles before a change in starting materials is allowed, but from there it depends if the slope falls below the threshold. This is further clarified in the figure of the SI section 5.3 which was also references in the revised manuscript.

- Page 12 – “With minerals used from the exact same origin” – it is already page 12, the minerals have been mentioned several times and still no mention of what they actually are

The specific mineral information was added to the MS. To ensure the same origin. the minerals were all homogenized from the same initial sample to reduce inhomogeneity in the natural material before washing.

- Page 12 – “With an experiment based on hundreds of repeated cycles” – this is not really the case here, is it?

This has been reworded into “several repeated cycles” in the revised manuscript.

- Page 12 – “Complex chemical mixtures with several different product species are increasingly leading to an analytical problem and the well-established way of analyzing every single product species of a MS spectrum is not feasible” – analyzing “several different product species” by MS methods is definitely feasible. Please rephrase.

We thank the reviewer for catching this mistake. This has been changed to read “Complex chemical mixtures with thousands to tens of thousands different product species are often seen as an analytical problem, and the well-established way of analyzing every single product species of a MS spectrum is not feasible.”

- Page 12 – “This is the reason that researchers attempt to develop a more system level approach, to look into changes, functions and trends in spectra instead of attempting to identify isolated product species” – if there are indeed other attempts, references are needed. I am sure this is known to the authors but to analyse a complex mixture one need not isolate the species – metabolomics approaches fare quite well here (e.g. Keller, M. A., Turchyn, A. V. & Ralser, M. Non-enzymatic glycolysis and pentose phosphate pathway-like reactions in a plausible Archean ocean. Mol Syst Biol 10, (2014).)

We have added the required references to the revised manuscript. The field of metabolomics is indeed advancing heavily, but a metabolomic search requires an appropriate library which currently is mostly available solely for biomolecules, or at least known molecule classes. As we tried to come up with an open experimental approach, looking for any kind of molecule, a metabolomic search was not feasible in this experimental design.

- Page 12/13 – *“Previous recursive chemistry experiments already raised the problem of addressing every single feature of an experiment” – references to these previous experiments are needed*

References to these experiments have been added to the revised manuscript.

- Page 13 – *“In our experiment, we are aiming for a rise in the complexity or information content of our input molecules, looking for an increase in the mass of the product species in addition to the use of the algorithm.” – this is a great way to state the aim of this study but it should appear on page 1, and not page 13.*

We agree with the reviewer and thank them for pointing this out. This statement is now made earlier in the revised manuscript.

- Page 14 – *“This observation can be related to the use of minerals. The minerals we used have been new and washed for each run individually” – the fact the minerals were used fresh and prewashed has been stated several times in the text already, but the identity of these minerals is (still!) unknown*

The minerals used has now been explicitly stated in the method section of the revised manuscript.

- Page 15 – *“platform for artificial life experiments” – throughout the paper I was convinced the application of the platform was in prebiotic chemistry/origin of life experiments. Artificial life is quite a different domain, and it cannot be used interchangeably with prebiotic chemistry. One could compare the origin of life research, conceptually, to an interpolation between geochemistry and biology, while artificial life is essentially an extrapolation.*

We thank the reviewer for pointing out this mistake which has been corrected in the revised manuscript.

- Page 15 (bottom) – *“while not generating the chemical mess” – any idea why? In an “unconstrained multicomponent system”, what provides this constraint?*

By chemical mess we mean unconstrained combinatorial explosion. The Mass Index metric falls to lower values in the case of a combinatorial explosion, so the fact that the algorithm successfully steers the experiment towards rising Mass Index, shows that this state has been avoided, see: Doran *et al.*, Emergence of Function and Selection from Recursively Programmed Polymerisation Reactions in Mineral Environments. *Angewandte Chemie International Edition* **58**, 11253-11256 (2019).

- Page 15 (bottom)/Page 16 (top) – *“most of the heaviest species produced are in the zone of a mass around 240 m/z”, “We achieved the performance of recursive experiments generating up to 28602 unique product species in a single cycle” – what does this mean for the bigger picture? What are the implications?*

The finding that the majority of the species found in that zone can be caused by several different effects. It might be based of the selection of solely small molecule building blocks, or the stability of larger molecules in solution. Another important effect which needs to be accounted for is as well, that some larger molecules are more difficult to ionise in the MS which can further lead to the fact that larger molecules are less likely to be detected. It would be interesting to investigate in the future, what changes need to be made to the experiment to produce molecules larger than 240 m/z.

The value for the species count was incorrect as there was a fault in the thresholding of the used code to analyse the sample and we have corrected the number, which is 5256 in the revised manuscript as well as we added a detailed description of the thresholding to section 5.4 in the SI.

- Page 17 – Mineral wash workflow – again, what are these minerals? In fact, I cannot find this information in the SI either. Minerals like FeS₂, MnO₂ or SiO₂ (off the top of my head) have strikingly different properties and could dramatically change the chemical landscape in some cases.

This has now been corrected in the revised manuscript.

- SI – a lot of information/data is present here, referring to which in the main text could help build the narrative of the paper. Why not use it in the revised version?

Several references to the SI have now been added to the revised manuscript.

I believe this paper has lots of potential. There is plenty of interesting information and useful conclusions that it could teach to the reader, but these conclusions are either indirect and buried in the text or not stated explicitly at all.

We thank the reviewer for their careful and critical reading of the manuscript, which has given us several pointers for improving the work. We have revised the manuscript and SI extensively to address these comments.

The fact that adding random chemicals to a reaction mixture, repeatedly, leads to a (sometimes less, sometimes more) complex mixture of products is not a revolutionary observation. Complexity for the sake of complexity cannot be the goal here, can it?

Actually, partly yes, if the highest m/z would increase as a function of the cycle, was the hypothesis to be tested. The metric used is a proof of concept for the use of a metric to measure the entire complexity of a product mixture through LC-MS. While the idea of doing this, and the obvious usefulness has been widely discussed, there are few examples where it is attempted.

Alternatively, is the experimental setup the major focus?

To some extent, yes. Using automation for prebiotic chemistry is also widely discussed, the field still lacks any sort of useful systems of that sort. This work demonstrates that automated platforms can be developed by prebiotic chemists, exploring dynamic unconstrained mixtures algorithmically with limited human intervention or bias.

There are lots of observations made in the text but not nearly enough interpretation. To give an example, on page 14 it is stated “we see that the mass index can increase even when there is not a dramatic increase of the mass of the heaviest product in the cycle” – and this is it. No discussion/interpretation of this observation, even speculatively. Instead, what follows is the Conclusions section, which begins by restating what was done in this study.

We thank the reviewer for highlighting these issues. We expanded and sharpened the discussion in the revised manuscript.

This paper requires major revisions.

Please articulate directly the following in the text: what exactly does this study teach us? what is the main point? how can the message be used by prebiotic chemists to improve their approach to experiments?

We have expanded our discussion in the second to last paragraph to read “The algorithm controlled and adjusted the experimental conditions (composition of the feedstock) on the fly by using an automated decision-making metric, which enabled the computer to interpret the MS data and make conclusions using data from previously executed cycles. Thanks to this feature, this system could be used by other scientists to explore the expansion of the chemical reaction network starting from simple organic compounds to include more complex molecules, adapting the feedstock based on the identification of key chemical species. Such

an experiment could be used to test long standing hypotheses about the emergence biochemical pathways before enzymes. “

And in the last paragraph “A platform, like the one described here could be used to search for increasing complexity by which simple molecules become complex chemical systems. By identifying those processes which lead to the robust complexification of chemical mixtures could one day lead to the missing link for the chemical to biological transition.”

We hope these additions highlight just two areas where this approach might be used by the wider community.

Taking all the above into consideration, I believe a major revision is required before this manuscript can be published in Nature Communications.

We thank the reviewer for the suggestions made to the manuscript and believe that the revised version has very much improved.

Reviewer #4 (Remarks to the Author):

This communication describes an automated system for running long term chemistry experiments (days to weeks or months). It uses automated delivery of three chemical building blocks / reagents, chosen from a human preselected set of 18 low molecular weight molecules. After a set reaction time 70% of the reaction mixture is removed for analysis and replenished with building blocks to recursively drive the system out of equilibrium on a cycle by cycle basis. A decision algorithm takes MS data from the end of each cycle and compares molecular diversity and higher order molecular weight species using a “mass index” metric to compare one cycle to another. It automatically changes the reagent set delivered according to the rate of change in the mass index. The goal is to over time drive the system to higher molecular weight species as a model of prebiotic chemistry.

This is an excellent summary of our work. The only clarification is that based on our metric, the Mass Index, we aimed not just to drive the system to higher molecular weight species but towards states with fewer species overall.

As such this is a novel approach to achieve the stated goal of creating a platform for “designing new types of origin of life experiments that allow testable hypotheses for the emergence of life from prebiotic chemistry”.

We are very happy that the reviewer agrees with this approach being necessary in the field.

The mass index as a measure of bias toward higher order molecular species is the primary metric used to measure the success of experimental runs. The authors show several long-term runs where the mass index changes over the course of multiple cycles and in some cases demonstrates a clear shift to higher order molecular species. Perhaps not surprisingly given the chemical complexity of the reaction milieu, repeat of identical runs shown in Figure 5 seem to indicate different mass index results over the course of each run, which the authors explain as possibly due to differences in the physical structure of the minerals used in the reactor.

Yes, there are differences in the mass indices shown in figure 5 but there are consistent peaks which appear in the higher resolution mass spectra from the offline analysis. The re-occurrence of those peaks suggests that significant portions of the mixtures were reproduced from run to run but there are variances which emerged likely due to the specific minerals used.

While a laudable approach to start to address this highly complex and burgeoning field in prebiotic chemistry (and perhaps a new arm of chemistry in general), it raises more questions than it provides answers about the kind of data it can produce and how to analyze it.

We are glad the review found our work thought provoking and we hope we have revised the manuscript clarify their questions about the results presented here.

The automated experimental approach is a novel method for handling long range chemistry but is largely missing controls or standards by which to measure and compare data. For example, some kind of control experiment (perhaps through a set of baseline runs, or e.g. like the non-recursive experiment example reported in Doran et. al. Angew. Chem. Int. Ed. 2019, 58, 11253–11256) could help to define just how different the repeat identical runs in Figure 5 are. This may also enable comparison of different run conditions, at least at a systems level. This may be planned for future exploration using the described set up, but some discussion on how the authors might go about this is warranted.

We are glad that the reviewer agrees that the approach is interesting. We added data of mineral control experiment to section 4.1.3 of the SI. These experiments compare recursive runs with and without minerals as well as analysed water stirred with minerals. The base peak chromatograms of these runs showed some peaks, but these peaks were consistent throughout 168 hours of stirring and heating. This shows that the Mass Index would not be changed throughout a prolonged experiment by the use of minerals, excluding the initial wash out reported.

Along these lines, some more detailed discussion on how this platform would be used in future to design experiments around Origins of Life, prebiotic chemistry, or just long range chemistry would help to explain the value of the system.

We have expanded our discussion in the second to last paragraph to read “The algorithm controlled and adjusted the experimental conditions (composition of the feedstock) on the fly by using an automated decision-making metric, which enabled the computer to interpret the MS data and make conclusions using data from previously executed cycles. Thanks to this feature, this system could be used by other scientists to explore the expansion of the chemical reaction network starting from simple organic compounds to include more complex molecules, adapting the feedstock based on the identification of key chemical species. Such an experiment could be used, for examples, to test long standing hypotheses about the emergence biochemical pathways before enzymes.”

And in the last paragraph “A platform, like the one described here could be used to search for increasing complexity by which simple molecules become complex chemical systems. By identifying those processes which lead to the robust complexification of chemical mixtures could one day lead to the missing link for the chemical to biological transition.”

We hope these additions highlight just two areas where this approach might be used by the wider community.

Comments on writing style: While an interesting paper, the writing is very poor and overly complex at times making this a very difficult concept to get across to readers unfamiliar with prior work. It needs significant editing for English and readability.

The language in the manuscript has been significantly revised and the readability improved.

Secondly the principles of how the system works are difficult to understand without looking at the Supporting Information. It is recommended that a perhaps more detailed version of Figure 19 in the SI be brought into the main manuscript as a way to graphically explain the experimental design and automated decision-making workflow.

We thank the reviewer for the suggestion and have referenced the SI sections more explicitly in the revised manuscript. For the clarity of the main manuscript we prefer to keep this rather technical figure in the SI.

Full description of all of the runs shown in Figure 4 is missing. Runs B,C and D would appear to be identical runs if these are the same as described in the SI but this is not referenced or discussed in the text. How do these relate to runs 10,11 and 12 in Figure 5?

We thank the reviewer for pointing out the confusing labelling of the runs. Run B, C and D are indeed identical to run 10, 11 and 12. The run description in the SI has now been updated, showing how each run is labelled and which runs relate to each other.

Notably the mass index for run 10 is the highest observed in this paper, yet not discussed.

We have now added a discussion of that observation to the revised manuscript which reads “The highest observed Mass Index was in run A with 9.63 and 5029 unique product species. This is interesting as that amount of product species is on the higher range of the observed count. This means that in this particular cycle, the mass of the heaviest product was been so high that the Mass Index was calculated high even with that number of species. This can lead to the conclusion that in this cycle, the randomly chosen input set lead to a increased complexity of the product mixture.”

The minerals used in the described experiments are not listed in the experimental or SI.

This has now been corrected in the revised manuscript. The minerals and the preparation details are described in the Methods section of the manuscript and section 2.2 of the SI.

There is some variance in nomenclature for the “mass index” in the main manuscript and “m/z index” in the SI which is potentially confusing.

We thank the reviewer for noticing this and it has been corrected in the revised SI to refer to Mass Index everywhere.

Reviewers' Comments:

Reviewer #1:

Remarks to the Author:

I see improvements in the manuscript (especially: language, more explanation especially in the SI). There were no changes in the underlying setup or experiments (as they would probably need several months of work). For me this work still contains too many blockers for publication in Nature Communications, and I stand by the comments I made in the initial round of review. It is true that the experimental details are very-well explained and detailed. This, however, to me, is not sufficient for acceptance; the science using the experimental set-up does not convince me.

First, my comments related to the authors' rebuttal.

- "Our goal is not to reproduce the [...] and we hope our clarifications below convince the reviewer."

I understand that this scope of this work is also to demonstrate the system, not only potential applications related to the emergence of life. I understand this, but even taking it into account I am not convinced that this work meets the standard of Nature Communications yet.

- "... that urgent publication is needed". The authors should not put pressure on the peer-review process to force a rapid publication. If they want other researchers to see their work, nothing prevents them from publishing it on a preprint server. This is accepted by many journals now.

- Dilution issue. I agree that molecules binding to the mineral surfaces will not be diluted. For the rest, I do not agree with the authors. The authors say "The critical exception to this is any compounds which are amplified in the reaction cycle", and use this argument to justify many of the choices and potential problems of their work. To me, this is plain wrong, and I fear that the authors lack some mathematical understanding. All the molecules (except the ones bound to the minerals) are equally distributed in the liquid. When taking away 70% of the liquid, 70% of any molecule in solution will disappear from the solution. After replenishing, the only molecules that will not be diluted are the ones that are present in the same or higher concentration in the added replenishing liquid (meaning: one of the 18 starting materials). All the other molecules **will be diluted**. There is no amplification. Some of the initially present molecules will react to something else. I will give a few examples to make this clear.

A) at the end of a cycle, the solution contains 1 mol/l of molecule X (we assume that from the compounds added in the next cycles, it is impossible to create X again). 70% is removed and replenished with 1 mol/l of molecule Y. Now, before anything reacts, there is 0.3 mol/l of X and 0.7 mol/l of molecule Y. If they react together, $X+Y \rightarrow Z$, there can be at most 0.3 mol/l of Z. At the beginning of the next cycle, there will be 0.09 mol/l of Z. No amplification! It is impossible for anything based on molecule X not to be diluted.

B) Let's consider that in a cycle Potassium pyrophosphate was added, and in the subsequent cycles no phosphorus is added at all. At the end of that cycle, let's say there are exactly one mol of phosphorus atoms. The phosphorus atoms will be equally distributed in the solution at any point in time (again excluding the minerals). This means that at each cycle, 70% of the phosphorus atoms will disappear. There is, mathematically speaking, no possibility for amplification. After ten cycles, there **will be** only $6 \cdot 10^{-6}$ mol of potassium atoms. In this example I chose to speak of an atom, but the same applies to molecules containing those atoms.

In summary: despite the statements about amplification by the authors, there is no such thing. Dilution will happen.

- In view of this dilution issue, I stand to what I said in the initial review about the "moving window of reactivity", and that at cycle 150, one basically sees only what happened in the last 10 cycles.

- Mass index. I agree that thresholding mass spectra is common in analysis. However, thresholding becomes problematic when dilution happens, as I explained in the first round of review. In my eyes, this formula is not stable enough for an iterative setup such as the one in this work. Even the authors acknowledge that the mass index is a "sensitive system". To me, an

adequate metric should reduce the noise of the underlying data, not amplify it. I would have the same opinion regarding the previous work of the authors in Doran et al. 2019.

- Reproducibility. The authors reject my comment about reproducibility and say that the attempt at reproducibility did not fail. I think it is important to differentiate between operation reproducibility (the machine does the same operations) and result reproducibility. The results are clearly not reproducible, as is shown by the different evolution of the Mass Index. Furthermore, this issue is exacerbated by the fact it is the very metric doing the decisions in a normal run which is not reproducible.

- Number of product species. I appreciate that the authors tried to find another way to calculate the number of product species to reduce the unrealistic number of above 20k, but the new way of setting the metric does not convince me, as I do not expect a comparison of the resulting numbers of product species to be meaningful for different solutions. The threshold they set depends on the peaks in the spectrum and their intensities. If I start with a solution and use their formula to get the number of compounds and get, let's say, 5000. If I then add a compound in powder form, so that it will have one or several very high-intensity peaks, this will result in an increase of the threshold - which may result in, let's say, 4000 species. So by adding a compound in powder form, without diluting any of the other compounds, I decreased the number of peaks from 5000 to 4000. This feels wrong. Also, I still struggle to understand the real meaning of those numbers (if they have any).

- "The data is complementing itself". I still don't understand this sentence, not sure if this is correct English.

Second, new comments related to the new manuscript version (or additional comments for things in both versions):

- "unconstrained": unless I missed it, the authors should define what "unconstrained" means in the context of this work.

- "[...] running a continuous experiment with several hundred cycles for more than 30 days without human intervention.": This is still not adequate; a few sentences later the authors mention the manual changing of the vials every three days. So the sentence should say: "almost without human intervention".

- Figure 5a). I don't understand why this figure needs to be in two dimensions: unless I am mistaken, all the points will *by construction* lie on the diagonal. Therefore, it is hard to see the meaning of that Figure, and also this does not support the argument of reproducibility. What the authors probably intended to convey was the distribution (histogram) of m/z values for the runs 10, 11, and 12, compared to the other runs? I feel like this would be a more adequate figure.

Reviewer #2:

Remarks to the Author:

I have carefully read through the Reviewer Comments and the Author Responses to all four reviews. For my comments, I am satisfied with the responses and changes made to the manuscript. Although I will not comment on the responses made to the other reviewer comments, I will say that I felt that all reviewer comments were reasonable and accurate.

I have read through the revised manuscript and feel that it is much improved from the prior submission. There are only two minor changes that I would suggest.

1. The last sentence of the abstract reads "This approach is an important step in designing new types of Origin of Life experiments that allow testable hypotheses for the emergence of life from prebiotic chemistry." Perhaps a personal preference, but I do not like to read in papers that the

authors believe their work to be 'important'. We should all be working on what we believe to be important problems, but only time really tells whether or not a particular piece of work is truly important. I suggest changing this sentence to "This approach represents what we believe to be a necessary step towards enabling the types of experiments that are now needed for testing chemical hypotheses for the emergence of life from prebiotic chemistry."

2. In Figure 3 the phrase "reported in Doran's paper" is rather colloquial. I suggest changing to the more standard "reported by Doran et al."

Reviewer #3:

Remarks to the Author:

The revised manuscript reads significantly better than before. The language has been improved, the clarity – increased, and the assumptions and goals are now much more explicitly stated. Terms and concepts typical for the authors' own discipline are now explained (e.g. the molecular transformer) which has further improved the accessibility of the paper.

The way I understand the purpose of this manuscript now is that it is meant to illustrate a setup where each of the variables can be easily changed, and it is not the specific compounds that are the objective, but rather pinpointing global trends (or the lack of them in some cases) that may emerge from a multi-cycle experiment. The idea of making the setup design and the code open access is commendable and this is something that should be highlighted here instead of the prebiotic applicability of this concrete choice of parameters (which, as stated above, I understand is an illustration of how the system works and what possible outcomes may be obtained rather than these specific outcomes being applicable to the origin of life on Earth). It needs to be stated absolutely clearly and unambiguously that the goal is not to identify novel chemical outcomes but to propose a technological development (i.e. the robotic platform) and the specific chemical outcomes are an illustration of the platform's performance (and not the goal in itself). This may sound redundant but I do believe making this point absolutely clear in the manuscript will be crucial for the future reception of this paper within the prebiotic chemistry community, which largely tends to focus on individual reactions and intermediates instead of global trends in large systems.

Regarding the point raised by another referee about Figure 5: I am wondering whether it would be useful to calculate how much the mass index changes per unit mass change of the threshold? Would this be an informative metric?

I still have a slight problem with Figure 1. Structures in Figure 1a) are illegible, especially in the inner circle (light blue). Not sure how to understand Fig 1b), without knowing what the nodes correspond to. Some explanation is necessary, either in the main text or in the caption, or in both. Perhaps stacking Figure 1a) and b) will help.

One more thing I find hard to understand in the author's rebuttal is "The aim was to solely concentrate on small functional building block molecules with no immediate connection or function". How can something be functional and at the same time have no immediate function? What am I missing here?

My previous comment concerning the lack of information about the type and characteristics of the minerals used has been addressed, and controls with/without minerals have been added, which has improved the scientific quality of the paper.

It is also great the authors attempted to identify the products. What would be even better, still, is some attempt to rationalise what this means, apart from the increase in complexity. Is one type of reactions prevalent over the others, for example? It is interesting that the median of all m/z values of all cycles is under 240 m/z. From the origin of life perspective, that would mean (correct?) a complexity level of molecules anywhere between e.g. citric acid (192 Da) or glucose (180 Da) and e.g. adenosine (267 Da), with more complex structures being arguably quite rare.

The term "possible reaction" is still not explained – and it is important to know whether it is based on thermodynamics or some other parameter.

Minor comment: P4 (top) – aim/aimed/aiming is now used in almost every sentence, but this is easy to fix.

Reviewer #4:

Remarks to the Author:

This original technical comments in the first assessment by this reviewer have been substantially addressed in this revision. The clarity and consistency of explanation of the scientific work as well as intent of the publication are much more clear and understandable.

While the introduction and conclusion sections have been appropriately edited for a native English writing style, the results and discussion sections still have multiple errors and awkward phrasing that need to be reviewed by a native English speaker prior to publication in a journal such as Nature Communications. This is required in the main text and figure legends.

Reply to reviewer comments. Reviewer comments in italics, our reply in normal type.

REVIEWER COMMENTS

Reviewer #4 (Remarks to the Author):

This original technical comments in the first assessment by this reviewer have been substantially addressed in this revision. The clarity and consistency of explanation of the scientific work as well as intent of the publication are much more clear and understandable.

We thank the reviewer for their comments and are glad the reviewer approves of the revisions made and agree that the revision helped to make the work clearer.

While the introduction and conclusion sections have been appropriately edited for a native English writing style, the results and discussion sections still have multiple errors and awkward phrasing that need to be reviewed by a native English speaker prior to publication in a journal such as Nature Communications. This is required in the main text and figure legends.

We thank the reviewer for this input, the revised manuscript, figure captions and SI have been proofread by multiple native speakers, and we hope that we have addressed all the grammatical errors.

Reviewer #3 (Remarks to the Author):

The revised manuscript reads significantly better than before. The language has been improved, the clarity – increased, and the assumptions and goals are now much more explicitly stated. Terms and concepts typical for the authors' own discipline are now explained (e.g. the molecular transformer) which has further improved the accessibility of the paper.

We thank the reviewer for their suggestions and are glad the reviewer thinks that the revised manuscript has been improved in terms of language, clarity, assumptions and goals as well as accessibility.

The way I understand the purpose of this manuscript now is that it is meant to illustrate a setup where each of the variables can be easily changed, and it is not the specific compounds that are the objective, but rather pinpointing global trends (or the lack of them in some cases) that may emerge from a multi-cycle experiment.

This is correct, we aimed to emphasize the development of the platform and have provided the results of a first set of experiments, and we hope that others in the community will use this approach and technology to devise many more experiments.

The idea of making the setup design and the code open access is commendable and this is something that should be highlighted here instead of the prebiotic applicability of this concrete choice of parameters (which, as stated above, I understand is an illustration of how the system works and what possible outcomes may be obtained rather than these specific outcomes being applicable to the origin of life on Earth). It needs to be stated absolutely clearly and unambiguously that the goal is not to identify novel chemical outcomes but to propose a technological development (i.e. the robotic platform) and the specific chemical outcomes are an illustration of the platform's performance (and not the goal in itself). This may sound redundant but I do believe making this point absolutely clear in the manuscript will be crucial for the future reception of this paper within the prebiotic chemistry community, which largely tends to focus on individual reactions and intermediates instead of global trends in large systems.

We thank the reviewer for this suggestion and totally agree. To make this clear we have added the following sentence to our conclusion: "Therefore, we hope that others will adopt our approach described here, and in the SI, so a common experimental standard can be adopted for these types of experiments. We hope this will enable the development of a new global effort for a 'big-data' origins of life search experiments.."

Regarding the point raised by another referee about Figure 5: I am wondering whether it would be useful to calculate how much the mass index changes per unit mass change of the threshold? Would this be an informative metric?

This is a good idea, and we have added this as Figure 26 in the SI. Here we show the Mass Index calculated for all the samples in Run 10 evaluated using different noise thresholds. Each line is one sample from Run 10. If the threshold is set too low, the mass index is dominated by peaks from detector noise and this is removed around 10^5 - 10^6 , resulting in the increase in the mass index (because the number of total peaks is greatly reduced by removing noise). Setting the threshold too high $\sim 10^7$, reduced the mass index by filtering true signal peaks. The different colors represent different samples while the black vertical dashed line indicates the threshold used during the experiments.

I still have a slight problem with Figure 1. Structures in Figure 1a) are illegible, especially in the inner circle (light blue). Not sure how to understand Fig 1b), without knowing what the nodes correspond to. Some explanation is necessary, either in the main text or in the caption, or in both. Perhaps stacking Figure 1a) and b) will help.

We thank the reviewer for this suggestion. Figure 1 has now been stacked and further explained in the caption in the revised manuscript.

One more thing I find hard to understand in the author's rebuttal is "The aim was to solely concentrate on small functional building block molecules with no immediate connection or function". How can something be functional and at the same time have no immediate function? What am I missing here?

With this sentence we tried to explain that we selected reactive small building blocks but selected these without targeting specific known pathways or reactions, as to avoid an "immediate function" of the specific building block. We meant to say we meant concentrate on small reactive building blocks that were not designed with a specific reaction in mind.

My previous comment concerning the lack of information about the type and characteristics of the minerals used has been addressed, and controls with/without minerals have been added, which has improved the scientific quality of the paper.

We are glad the reviewer approves of the additional information presented and sees an improvement of the scientific quality of the paper.

It is also great the authors attempted to identify the products. What would be even better, still, is some attempt to rationalise what this means, apart from the increase in complexity. Is one type of reactions prevalent over the others, for example? It is interesting that the median of all m/z values of all cycles is under 240 m/z . From the origin of life perspective, that would mean (correct?) a complexity level of molecules

anywhere between e.g. citric acid (192 Da) or glucose (180 Da) and e.g. adenosine (267 Da), with more complex structures being arguably quite rare.

As it is shown in the SI section 4.2.2, the 2206 known predicted reactions could have resulted in up to 429 possible products of which we found 22 matches in our samples. These products are still very different and it is not possible to group them in a logical way, but it was interesting to see that the highest matched product species had a molecular weight of 175.06 (3-Oxetanone) which lies under the 240 m/z threshold.

We agree that the 240 m/z threshold is a curious finding but further experiments with alternative techniques would be necessary to further confirm the number of 240 m/z as a barrier, as, as mentioned in the manuscript, factors like magnetic stirring and the use of ESI-MS can also break down bigger molecules, leading to a lower detection of those.

The term "possible reaction" is still not explained – and it is important to know whether it is based on thermodynamics or some other parameter.

This has now been added to the revised manuscript. A possible reaction in this context is solely a reaction predicted by the molecular transformer. We have clarified this in the text to say "possible reactions, which means they can be predicted based on the structure of the reactants and previously reported reactions from the literature."

Minor comment: P4 (top) – aim/aimed/aiming is now used in almost every sentence, but this is easy to fix.

This has now been corrected in the revised manuscript.

Reviewer #2 (Remarks to the Author):

I have carefully read through the Reviewer Comments and the Author Responses to all four reviews. For my comments, I am satisfied with the responses and changes made to the manuscript. Although I will not comment on the responses made to the other reviewer comments, I will say that I felt that all reviewer comments were reasonable and accurate.

We thank the reviewer for their comments and suggestions and are glad that we have been able to change the manuscript to their satisfaction.

I have read through the revised manuscript and feel that it is much improved from the prior submission. There are only two minor changes that I would suggest.

1. The last sentence of the abstract reads "This approach is an important step in designing new types of Origin of Life experiments that allow testable hypotheses for the emergence of life from prebiotic chemistry." Perhaps a personal preference, but I do not like to read in papers that the authors believe their work to be 'important'. We should all be working on what we believe to be important problems, but only time really tells whether or not a particular piece of work is truly important. I suggest changing this sentence to "This approach represents what we believe to be a necessary step towards enabling the types of experiments that are now needed for testing chemical hypotheses for the emergence of life from prebiotic chemistry."

We thank the reviewer for this suggestion and the sentence has now been changed in the revised manuscript.

2. In Figure 3 the phrase "reported in Doran's paper" is rather colloquial. I suggest changing to the more standard "reported by Doran et al."

This has now been corrected in the revised manuscript.

Reviewer #1 (Remarks to the Author):

I see improvements in the manuscript (especially: language, more explanation especially in the SI).

We are glad that reviewer 1 thinks the manuscript is improved.

There were no changes in the underlying setup or experiments (as they would probably need several months of work).

This is correct, but also it is not clear what additional hypothesis is to be tested. We believe that our central thesis has been shown to be testable and that the data is both interesting and important. Also the fact that our system can be easily replicated by others is a key aspect.

For me this work still contains too many blockers for publication in Nature Communications, and I stand by the comments I made in the initial round of review.

We do not understand this comment, especially by what is meant by the term 'blockers'. We have demonstrated that the design of experiments works, and that the trends are reproducible. We have purposely been as open as possible.

It is true that the experimental details are very-well explained and detailed. This, however, to me, is not sufficient for acceptance; the science using the experimental set-up does not convince me.

This comment contrasts with the other referee reports, and importantly at no point has the referee been able to give specific reasons for why the setup is not convincing. The central thesis was to explore 'a robotic prebiotic chemist equipped with an automatic sensor system designed for long-term chemical experiments exploring unconstrained multicomponent reactions, which can run autonomously over long periods and uses only simple chemical inputs.' Given that is exactly what we did, and the referee does not seem to disagree with the central idea, we struggle to understand their problem. At the simplest level the thesis was to allow long term experiments to be done, and to be able to address reactions and conditions covered by other origin of life experiments. Investigation of the literature literally reveals hundreds of manual experiments all with different set ups and slightly different process conditions that could easily be unified using this approach.

This means our system can be used to standardize the data collection. Such approaches are need for big data science today and this area is particularly in need since we are aiming to explore the idea that there might be a sensitive dependence on initial conditions, and this is not testable without such a set up. This is the critical aspect of searching for the emergence of chemical selection. This paper should be viewed as the very first attempt to provide a systematic approach that will allow others to build on and use the scientific method.

First, my comments related to the authors' rebuttal.

- "Our goal is not to reproduce the [...] and we hope our clarifications below convince the reviewer." I understand that this scope of this work is also to demonstrate the system, not only potential applications related to the emergence of life. I understand this, but even taking it into account I am not convinced that this works meets the standard of Nature Communications yet.

Since the reviewer does not give any further reasons we are not able to comment much other than point to the other reviewer comments, reiterate that our platform can do lots of origin of life type experiments described in the literature, but allows them to be standardized.

- "... that urgent publication is needed". The authors should not put pressure on the peer-review process to force a rapid publication. If they want other researchers to see their work, nothing prevents them from publishing it on a preprint server. This is accepted by many journals now.

This is highly perplexing, why should we not want a rapid publication? Many groups around the world are doing manual origin of life experiments. Many are not fully controlled, many require lots of person hours that could be avoided with our approach as well as have increased safety and accountability. Furthermore the

point of peer review is clear. Preprints are important for areas where there is competition of ideas. Here the work is highly technical and costly hence having a fully peer reviewed publication will greatly help uptake and confidence since the readers will know that it has been seriously scrutinized.

- Dilution issue. I agree that molecules binding to the mineral surfaces will not be diluted.

We are glad the reviewer acknowledges that compounds can be retained on mineral surfaces, this is important because it provides a mechanism to introduce a chemical history that is retained through many cycles, regardless of the subsequent dilution of the reaction mixture.

For the rest, I do not agree with the authors. The authors say "The critical exception to this is any compounds which are amplified in the reaction cycle", and use this argument to justify many of the choices and potential problems of their work. To me, this is plain wrong, and I fear that the authors lack some mathematical understanding.

There is no misunderstanding. The reviewer needs to think further than a simple kinetic model and although the reviewer thinks dilution will wash out any effects, our experiments show that there are persistent effects. Again, one of the key results is that changing the input reagents automatically did not decrease the Mass Index of the mixture over time, and in fact in many cases adding new reagents increased the Mass Index over consecutive cycles, suggesting the amplification of some species over the net dilution. This is incredibly interesting and requires extensive follow up with precise molecular level mechanistic investigations.

All the molecules (except the ones bound to the minerals) are equally distributed in the liquid. When taking away 70% of the liquid, 70% of any molecule in solution will disappear from the solution.

We agree that molecules in the solution will be removed equally, while those bound to minerals may persist for many cycles.

After replenishing, the only molecules that will not be diluted are the ones that are present in the same or higher concentration in the added replenishing liquid (meaning: one of the 18 starting materials).

This is true for the instant the new reagents are added to the reaction mixture. The subsequent reactions will change this.

*All the other molecules ****will be diluted****. There is no amplification.*

This is where we believe the reviewer has misunderstood the unconstrained nature of reaction mixtures. The argument the reviewer makes below are based on an understanding of single, isolated reactions from pure reagents. In the unconstrained mixtures studied here it is frequently the case that one product can be made from multiple reactions using different reagents, meaning that some products will be formed that persist through reactions cycles longer than others in the face of dilution. The mathematical argument the reviewer makes ignores this possibility, which is an assumption they do not justify, and has not been studied empirically. The analysis we did with the molecular transformer found that several combinations of reagents from our 18 inputs could react to yield identical products.

Some of the initially present molecules will react to something else. I will give a few examples to make this clear.

A) at the end of a cycle, the solution contains 1 mol/l of molecule X (we assume that from the compounds added in the next cycles, it is impossible to create X again). 70% is removed and replenished with 1 mol/l of molecule Y. Now, before anything reacts, there is 0.3 mol/l of X and 0.7 mol/l of molecule Y. If they react together, $X+Y \rightarrow Z$, there can be at most 0.3 mol/l of Z. At the beginning of the next cycle, there will be 0.09 mol/l of Z. No amplification! It is impossible for anything based on molecule X not to be diluted.

When specific products persist through many reaction cycle we say they have been amplified by the reaction. Again, the reviewer's analysis here ignores the possibility that Z may be formed from another reaction that does not depend on X or Y.

B) Let's consider that in a cycle Potassium pyrophosphate was added, and in the subsequent cycles no phosphorus is added at all. At the end of that cycle, let's say there are exactly one mol of phosphorus atoms. The phosphorus atoms will be equally distributed in the solution at any point in time (again excluding the minerals). This means that at each cycle, 70% of the phosphorus atoms will disappear.

We are interested in the persistence of products, not of input reagents. The question is therefore not about what will happen to the reagents upon serial dilution but rather what products forms and whether those products are formed robustly from the reaction mixture or only marginally. Those that are formed robustly (e.g. from multiple sets of reagent combinations) will persist (be amplified) in the face of dilution.

*There is, mathematically speaking, no possibility for amplification. After ten cycles, there ****will be**** only $6 \cdot 10^{-6}$ mol of potassium atoms. In this example I chose to speak of an atom, but the same applies to molecules containing those atoms.*

In summary: despite the statements about amplification by the authors, there is no such thing. Dilution will happen.

Our experiments showed that multiple product species could persist through reaction cycles at detectable concentrations. The reviewer has given an abstract mathematical argument that ignores key aspects of our experimental set-up. We hope we have illustrated to the reviewer why this argument is ill posed and actually the experimental data supports our view. So we would say that the fact our data shows that we are producing interesting compounds despite the possible effects of dilution is a key point and why the reviewer might concede this work should be published.

- In view of this dilution issue, I stand to what I said in the initial review about the "moving window of reactivity", and that at cycle 150, one basically sees only what happened in the last 10 cycles.

The reviewer has already agreed that compounds bound to mineral surfaces will persist through reaction cycles. Those compounds alone are sufficient to change the reaction conditions and persist for longer than 10 cycles. The three other reviewers have not taken issue with our description of the platform and our characterization of the experiments.

- Mass index. I agree that thresholding mass spectra is common in analysis. However, thresholding becomes problematic when dilution happens, as I explained in the first round of review. In my eyes, this formula is not stable enough for an iterative setup such as the one in this work.

We are glad the author understands that thresholding is a routine process in mass spectrometry, common to every analysis performed. Although this reviewer made no suggestions, one of the other reviewers did, and we have incorporated the suggestion from another reviewer and added a new figure to our SI, Figure 26, which shows how the mass index changes for the samples in Run 10 as the threshold is changed.

Even the authors acknowledge that the mass index is a "sensitive system". To me, an adequate metric should reduce the noise of the underlying data, not amplify it. I would have the same opinion regarding the previous work of the authors in Doran et al. 2019.

The mass index is sensitive in that removing a few peaks from the spectrum can cause a significant increase in the mass index. In the face of this sensitivity, we only included peaks above a specific threshold (which was constant through all the runs) to ensure that noise peaks due to detector noise were not included in the calculation. We have explored multiple different metrics in our SI (section 5). The reviewer has not suggested alternatives for us to test or analyze.

- *Reproducibility. The authors reject my comment about reproducibility and say that the attempt at reproducibility did not fail. I think it is important to differentiate between operation reproducibility (the machine does the same operations) and result reproducibility.*

We agree this distinction between operational reproducibility and the reproducibility of the product mixtures is essential. In our experiments, the platform executed identical unit operations, adding reagents of the same concentration in sequence. Our high-resolution analysis shows that many of the products from the runs were identical. In this sense the experiment is reproducible. Interestingly, there were some ions present in each run that were not observed in the others. Understanding how these unique products emerged out of identical experiments represents an exciting new avenue of research we hope to pursue with this platform.

The results are clearly not reproducible, as is shown by the different evolution of the Mass Index. Furthermore, this issue is exacerbated by the fact it is the very metric doing the decisions in a normal run which is not reproducible.

The trajectories of the mass index show that the chemical systems changed differently over the course of many cycles. Our high-resolution analysis of the product mixture showed the composition of those mixtures were very similar but had important differences. The Mass index for different mixtures are different. If we perform the mass index calculation on the same mixture we get the same value, which means the calculation is reproducible.

- *Number of product species. I appreciate that the authors tried to find another way to calculate the number of product species to reduce the unrealistic number of above 20k, but the new way of setting the metric does not convince me, as I do not expect a comparison of the resulting numbers of product species to be meaningful for different solutions. The threshold they set depends on the peaks in the spectrum and their intensities.*

This thresholding is based on the fact that majority of peaks in a high resolution spectrum will be due to detector noise (the very reason thresholding is so important in mass spectrometry). By assigning a threshold based on the median intensity, we derive a threshold that can adapt to the noise signal in the spectrum. Using the median ensures that the addition of a few high intensity peaks does not shift the threshold significantly (the median of a set of values is robust to extreme values, unlike the mean).

If I start with a solution and use their formula to get the number of compounds and get, let's say, 5000. If I then add a compound in powder form, so that it will have one or several very high-intensity peaks, this will result in an increase of the threshold - which may result in, let's say, 4000 species. So by adding a compound in powder form, without diluting any of the other compounds, I decreased the number of peaks from 5000 to 4000. This feels wrong.

This is not true, the median intensity peak will not increase significantly because adding a single (or a few peaks) will not shift the distribution significantly. In any case, it is not applicable to the experiments here because every compound used in the experiments was added as an aqueous solution of standardized concentration.

Also, I still struggle to understand the real meaning of those numbers (if they have any).

The number of the product species describes the number of unique ions found in one sample. This value is one way to compare different experiments by comparing how it changes with cycle number. This is especially interesting as the number of unique peaks (species) is a factor in the mass index value calculation.

- *"The data is complementing itself". I still don't understand this sentence, not sure if this is correct English.*

This has been corrected in the revised manuscript to say "Figure 5c reveals that many features with high m/z are unique to one run and that more such features are seen in the repeat runs. Given how the Mass Index is

calculated, with especially heavy masses being weighted higher, this helps explain the origin of differences seen in the repeat runs compared to the initial experiment.”

Second, new comments related to the new manuscript version (or additional comments for things in both versions): - "unconstrained": unless I missed it, the authors should define what "unconstrained" means in the context of this work.

The work of “Surman, A. J. *et al.* Environmental control programs the emergence of distinct functional ensembles from unconstrained chemical reactions. *Proc Natl Acad Sci U S A* **116**, 5387-5392, doi:10.1073/pnas.1813987116 (2019).” Was referenced with the first mention of this term, additionally we added a sentence for further clarification to the revised manuscript.

In the context of this work, in which we are consistent with the definition of Surman *et al.*, the system is not restricted in environmental factors as, in this case within the compound library, to change the input composition at any given time without human intervention/ constraint. We have clarified this in the text to say “unconstrained (which means in this context, the system is not restricted within the compound library, to change the input composition at any given time without human intervention/ constraint)”.

- "[...] running a continuous experiment with several hundred cycles for more than 30 days without human intervention.": This is still not adequate; a few sentences later the authors mention the manual changing of the vials every three days. So the sentence should say: "almost without human intervention".

This has been changed in the revised manuscript.

*- Figure 5a). I don't understand why this figure needs to be in two dimensions: unless I am mistaken, all the points will *by construction* lie on the diagonal. Therefore, it is hard to see the meaning of that Figure, and also this does not support the argument of reproducibility. What the authors probably intended to convey was the distribution (histogram) of m/z values for the runs 10, 11, and 12, compared to the other runs? I feel like this would be a more adequate figure.*

Figure 5a and c show different representations of the same underlying data. With the majority of the data in the m/z range of 0-250 we use figure 5c to show how many points are common through the different runs. We have incorporated the reviewer’s suggestion and made figure 5a into a histogram that shows the consistency between samples in the different runs.

Reviewers' Comments:

Reviewer #1:

Remarks to the Author:

As in the two previous rounds of review, I still see blockers in the manuscript, which are, in order of importance:

- 1) dilution issue
- 2) reproducibility of the results
- 3) suitability of the mass index

Although I described my concerns about those points in detail two times already, the authors failed to convince me that these are not shortcomings.

It seems to me that they avoid addressing the real issues in their rebuttal (details below), and hence to me these problems have not been solved or addressed properly in the revisions.

I will, for the third time, go more in detail about those points further below.

The three issues, which I am repeating since the first round of review, and to which I haven't yet had a satisfying answer to until now, are fundamental enough to question the real impact of the experimental framework introduced by the authors.

In particular, because of the dilution issue, it becomes irrelevant that experiments ran for several weeks, since any moiety in solution at any point in time will be diluted away after a dozen cycles.

To quote the Nature Communications website: "Papers published by the journal represent important advances of significance to specialists within each field", or "novel and important research study of high quality and of interest to that specific research community."

One cannot expect all good articles to be published in high-impact journals; only exceptional ones make it there.

I therefore hope that the authors will understand my opinion that their work does not make the cut for Nature Communications.

Let me restate, in a condensed fashion, my main concerns about points 1)-3). For completeness, I advise anyone reading this to also read my related comments in the two first rounds of review.

1) Dilution issue

Even taking into account the possibility that a product can be generated with different sets of reagents (which I expect will be the exception rather than the rule), products will be diluted. As soon as a given molecule cannot be generated from the compounds *currently being added*, its concentration can only decrease. And it will hence disappear exponentially in the next cycles. For this reason, I find it adequate to speak of a window of reactivity, which defeats the purpose of running such long experiments. The authors say that "the data supports [their] view", but Figure 4 seems to rather prove my point: the long-term trend of the mass index curve is as often flat, decreasing and increasing. To me, this clearly cannot be used as a proof to say that dilution is not a problem.

The authors' reply did not counter the fundamental issue about dilution and instead argument with edge cases (multiple combinations of reagents producing the same product, molecules binding to the surface). I tried to give understandable examples to illustrate why dilution is a problem, but the authors disregarded these examples because they are supposedly too simple or too abstract. In fact it should be the authors' role to illustrate why they think dilution does not result in a window of reactivity, not the other way around. After all that, I find it worrying that the authors even speak of amplification.

2) Reproducibility of the results

In the rebuttal, the authors insisted on the reproducibility of the setup, or the reproducibility of the calculation of the mass index for an identical solution, but they did not acknowledge clearly that the results are not reproducible. By this I explicitly mean that two runs doing the same operations in the same order lead to very different curves of the mass index. Taking Figure 5 as an example, the results would be reproducible if runs 11, 12 and 13 led to identical values. Which is, clearly, not the case.

3) Mass index

My view is best described by quoting part of my comment from the first round of review:

"I can only look at the formula, and notice that it is, in my opinion, not very robust: a little change in the MS intensity (either from one cycle to the next, or because of experimental noise) can lead to one species being above the threshold or not. Such a little change in intensity can therefore lead to a big jump in the value of the mass index because the mass range or the number of considered peaks will suddenly be different. This may be one of the reasons why the reproducibility test failed."

In the rebuttal, the authors disregarded my point that thresholding can especially risky when combined with the dilution issue: by keeping the threshold constant, from one cycle to the next some of the peaks will change from counting 100%, to counting 0%. This is what makes the metric so sensitive in my opinion. The authors reproach (two times) that I did not make suggestions regarding this, so here we go: an adaptive formula that ensures that 1) a peak is counted partially in n_counts depending on its intensity, to avoid the "all-or-nothing" of a hard threshold, and 2) an adaptive mass range where m_max and m_min are determined by averaging several peaks, maybe also applying some partial counts. This may end up not being adequate, but the formula would be more robust, and less of a "sensitive system" as the authors formulate it.

To additional things that the reviewers mentioned in the rebuttal:

- If other referees did not raise the same points as I does not mean that I am wrong. (It does not mean that I am right either). Hence it should not be used as an argument to dismiss my views.
- I have nothing against a rapid peer-review process. But when, in the first rebuttal, the authors wrote that "urgent publication is needed" and underlined it, it did feel like pressure.

Reviewer #3:

Remarks to the Author:

The authors have already addressed my previous comments in their last response, so I only comment on their response to Reviewer 1. I think R1 and the authors are talking about two different things, and overall, I find the author's response compelling.

What R1 says about products being diluted makes sense if it was a linear reaction or several reactions in a linear series. If one assumes each compound can be made by multiple reactions and these reactions do not have to be the same in every run, because in every run the input compounds can change, then it is possible that some compounds will start to dominate if they are a result of several of such processes while other compounds will not.

The authors are focusing on amplification, which they are defining as persistence of certain products despite cyclic dilutions, while R1 understands amplification as an increasing concentration of a product of a linear $A+B \rightarrow C$ reaction. Each product can be made as a result of several reactions occurring at the same time in a multibranching network. One thing the authors could do to clarify this is to include the discussion from their answers to R1 in the SI, for example (and write explicitly in the main text how they are using the term amplification, although they have somewhat done that already). Alternatively, they might want to choose a different word than "amplified".

Reply to reviewer comments. Reviewer comments in italics, our reply in normal type.

REVIEWER COMMENTS

Reviewer #1 (Remarks to the Author):

As in the two previous rounds of review, I still see blockers in the manuscript, which are, in order of importance:

1) dilution issue

2) reproducibility of the results

3) suitability of the mass index

Although I described my concerns about those points in detail two times already, the authors failed to convince me that these are not shortcomings.

We genuinely did make a very big effort (outlined below) and we are willing to try again. We note that all the other referees are very happy and supportive and think that our efforts have been appreciated.

It seems to me that they avoid addressing the real issues in their rebuttal (details below), and hence to me these problems have not been solved or addressed properly in the revisions.

I will, for the third time, go more in detail about those points further below.

We disagree that we have avoided addressing the real issues. To remind the reviewer, we have made the following changes to the manuscript and SI:

- Clarified in detail the nature of our experimental design and pointing out how we understand our results in the context of dilution.
- Modified Figure 5 to show the differences and similarities between repeat runs.
- Demonstrated how a specific run would change given different thresholding values (figure 28 in the SI)
- Added information about the chemical selection process and properties of each of the input compounds (section 2.1 in the SI).
- Provided examples of how the mz-metric works in specific cases (SI section 3).
- Manually identified product species for an example run and listed their tentative formula (SI section 4.2).
- Added control experiments, investigating the leaching of minerals in solution with ICP (section 2.3 of the SI) and the behavior of minerals in a recursive set up (section 4.1.3 of the SI).
- Added comparison of the mass index and other alternative algorithms (section 5.4 in the SI).

We will attempt to will address these issues in more detail.

The three issues, which I am repeating since the first round of review, and to which I haven't yet had a satisfying answer to until now, are fundamental enough to question the real impact of the experimental framework introduced by the authors.

In this response, as in our previous responses, our goal is to directly address the reviewer's concerns. We have carefully reviewed their comments on all three matters and have added text or additional analysis to the manuscript and SI to address them.

In particular, because of the dilution issue, it becomes irrelevant that experiments ran for several weeks, since any moiety in solution at any point in time will be diluted away after a dozen cycles.

We disagree with reviewer 1 here and think this is based on the misunderstanding reviewer 3 raised. We focus on build-up of product species which persist over the dilution - perhaps due to a combination of adsorption to the mineral surfaces and through the intrinsic dynamics of the complex reaction network. It is not the case that all product species are diluted away after a dozen cycles in our experiment, and our data shows this. We have added further analysis of the data (SI section 4.2.3) showing that species remain in the product mixture over a dozen cycles.

To quote the Nature Communications website: "Papers published by the journal represent important advances of significance to specialists within each field", or "novel and important research study of high quality and of interest to that specific research community."

One cannot expect all good articles to be published in high-impact journals; only exceptional ones make it there.

We note that all the other reviewers have recommended publication and wish to state again, as we have already explained in the previous response, this paper is the very first attempt to provide an automated prebiotic chemistry platform, for long-term unconstrained reactions. It is not only relevant for the exploration of long-term experiments but also in the ways to approach big data sets and automating chemical reaction processes. This work addresses key aspects which chemistry research is currently in need of, and provides accessible, open, code solutions for problems like reaction automation and algorithmic data approaches for further research in the field.

I therefore hope that the authors will understand my opinion that their work does not make the cut for Nature Communications.

We think the latest clarification can convince the reviewer, as we have with the three other reviewers, that our results represent a significant advance that is of interest to the readers of *Nature Communications*.

Let me restate, in a condensed fashion, my main concerns about points 1)-3). For completeness, I advise anyone reading this to also read my related comments in the two first rounds of review.

1) Dilution issue

*Even taking into account the possibility that a product can be generated with different sets of reagents (which I expect will be the exception rather than the rule), products will be diluted. As soon as a given molecule cannot be generated from the compounds *currently being added*, its concentration can only decrease.*

This is the key point. When we see species persist in the face of dilution, we have assumed they are being amplified by the system. However, we are not making assumptions about the mechanisms underlying the generation of these molecules, we are reporting our results in which we see some product ions persist through many cycles including changes in the input compounds. To try to make our findings clear, we have incorporated reviewer 3's suggestion to avoid the word 'amplification' and instead we have phrased our findings in terms of 'persistence.' We hope this, in combination with the new analysis we have provided below will clarify this issue.

And it will hence disappear exponentially in the next cycles.

This is not what we see in our data. We have added a section to the SI (4.2.3) in which we tracked a handful of product peaks through an example experiment (Run G). In this analysis we used the higher resolution data from the Bruker Maxis (LC-ESI-MS) and focused only on those ions which we have been able to assign formula, AND which were predicted by the network analysis. Therefore, this graph is an

illustration for the underlying trends in the entire dataset. The threshold used for this instrument was different than that used in the online analysis, because it is a different instrument with different specifications. The threshold was set to 100, which is consistent with our other analysis using this instrument.

The followed peaks have been product species previously predicted by the network. While we see that some products are disappearing through dilution (pale blue and green line) we see also species persisting through the whole run (purple and dark blue line). This shows that even with changing input compositions, some product species can persist through the whole experiment and are not disappearing exponentially.

For this reason, I find it adequate to speak of a window of reactivity, which defeats the purpose of running such long experiments.

In this study we were interested in the persistence of products, not of input reagents. If the experimental setup was linear and solely be based on periodically changing single input compounds, we accept every product would be diluted with time. As shown above, we have analytically shown that product species persist over many more cycles than would be expected in a linear experiment. We have added a section to the SI (2.1) to illustrate how dilution does not result in a 'window of reactivity' in our experimental concept. Furthermore, as the reviewer has previously agreed, the existence of mineral surfaces provides a mechanism to induce a long history by preserving compounds in the system through many cycles. There may be other mechanisms responsible for the persistence of the product ions we have observed but that is a topic for future research.

The authors say that "the data supports [their] view", but Figure 4 seems to rather prove my point: the long-term trend of the mass index curve is as often flat, decreasing and increasing. To me, this clearly cannot be used as a proof to say that dilution is not a problem.

As we have discussed in detail in the manuscript, figure 4 does not show an immediate trend in the presented experiments, but rather shows the diversity of possible outcomes which we observed. As

the reviewer correctly observed, the curve is flat but also decreasing and increasing, which does not give information of the dilution of the specific species. While this figure was not used as an example that dilution is not a problem, the fact that we see curves rising over input changes indicates that species may persist. A better proof to observe that dilution is not a problem is the example we gave in the response above with the figure of SI section 4.2.3.

The authors' reply did not counter the fundamental issue about dilution and instead argument with edge cases (multiple combinations of reagents producing the same product, molecules binding to the surface).

The critical question in our experiment is whether any product species can persist in an unconstrained non-linear system. We have identified species (shown above and in SI section 4.2.3) which do. Our explanation regarding how this is possible depends on identifying the differences between the design of our system and a simplified linear reaction network. In light of our data, we do not think that the ability of compounds to bind to mineral surfaces or for multiple combinations of reagents to produce the same product represent edge cases.

I tried to give understandable examples to illustrate why dilution is a problem, but the authors disregarded these examples because they are supposedly too simple or too abstract.

The examples the reviewer gave are solely limited to linear reaction systems, while for these systems dilution would be a problem, our recursive reaction system is constructed differently with the mineral surface providing space for product robustness and persistence over dilution. Our system has implementations which change the properties of the reaction system, allowing non-linear reactions.

In fact it should be the authors' role to illustrate why they think dilution does not result in a window of reactivity, not the other way around.

We have added a section to the SI (2.1) to illustrate how and why dilution does not result in a 'window of reactivity' in our experimental concept.

"In this study we are interested in the persistence of products, not of input reagents. If the experimental concept were linear and solely be based on periodically changing input compounds, every product would become diluted over time. However, our experimental concept is different, and with the inclusion of minerals, the reaction system is not linear. Over time, many of the components of the mixture will be diluted to infinitesimally low concentrations, but not all. Binding to mineral particles can lead to product build up on their surface which may then lead to species amplification, as well as a change of the species concentration in the reactor over time. Products in our system can be made from multiple reactions / input compositions, including the breakdown components of other product species, which can lead to some specific product species becoming prevalent, even under different input conditions. The critical question is therefore not about what will happen to the reagents upon serial dilution, but rather what products are formed and whether those products are formed robustly from the reaction mixture or only marginally. Those that are formed robustly (e.g. from multiple sets of reagent combinations) will persist (be amplified) in the face of dilution. When specific products persist through many reaction cycle, we say they have been amplified by the reaction, while all species that do not increase will be diluted out." We hope that this can help the reviewer understand why we think persistence over dilution is possible in our reaction system.

After all that, I find it worrying that the authors even speak of amplification.

Following the suggestion from reviewer 3, we have changed the sentence in the manuscript to "By recursively cycling and diluting the product mixture with the addition of further starting reagents, our experiments were designed to dilute out any product compounds which are not robust during the

reaction cycle. Thus, the only compounds remaining in significant abundance after many cycles would be those that were produced over the cycle, persisted over dilution, or were bound to the mineral surfaces.”

We hope that this illustrates the results we describe in a more accessible and clear way, without causing confusion using the term ‘amplification’. In this context that term was always defined as “the persistence of product species despite cyclic dilutions”, but we accept that it could have been interpreted differently. We have also added a sentence to conclusions to make this clear and also explain that amplification could be a possibility.

2) Reproducibility of the results

In the rebuttal, the authors insisted on the reproducibility of the setup, or the reproducibility of the calculation of the mass index for an identical solution, but they did not acknowledge clearly that the results are not reproducible. By this I explicitly mean that two runs doing the same operations in the same order lead to very different curves of the mass index. Taking Figure 5 as an example, the results would be reproducible if runs 11, 12 and 13 led to identical values. Which is, clearly, not the case.

We have added changes on page 14 to make it more clear that the mass-index curve is not identical: “We can see that these experiments, prepared under the same conditions, differ from each other when measured online and as shown in (Figure 5, b) the curves of Run 10 and 11 lead to different curves of the mass index.”

And also: “We further tested all collected samples from each run on a more sensitive MS instrument after all runs had been completed. Those results (Figure 5, c) show that many of the products from these runs were identical but also differences in the overall product distribution of the different runs, especially in higher mass ranges, where we see many features that are unique to each run.”

The mass-index curve is, as the reviewer correctly states, not reproduced, but we found identical products in all 3 runs, when tested on a high-res MS instrument. We hope that the reviewer sees the challenges of the reproducibility with the changes in the manuscript now clearly addressed.

3) Mass index

My view is best described by quoting part of my comment from the first round of review: "I can only look at the formula, and notice that it is, in my opinion, not very robust: a little change in the MS intensity (either from one cycle to the next, or because of experimental noise) can lead to one species being above the threshold or not. Such a little change in intensity can therefore lead to a big jump in the value of the mass index because the mass range or the number of considered peaks will suddenly be different. This may be one of the reasons why the reproducibility test failed." In the rebuttal, the authors disregarded my point that thresholding can be especially risky when combined with the dilution issue: by keeping the threshold constant, from one cycle to the next some of the peaks will change from counting 100%, to counting 0%.

We did not disregard this point. In our rebuttal we have taken care to explain the role that thresholding plays in our analysis and included additional figures in the SI (Figure 27 and 28) to demonstrate that our system was removing noise but not significant numbers of true peaks.

This is what makes the metric so sensitive in my opinion.

As above, we have shown in the SI (figure 27) that the metric is not overly sensitive to the threshold.

The authors reproach (two times) that I did not make suggestions regarding this, so here we go: an adaptive formula that ensures that 1) a peak is counted partially in n_counts depending on its intensity, to avoid the "all-or-nothing" of a hard threshold, and 2) an adaptive mass range where m_max and m_min are determined by averaging several peaks, maybe also applying some partial counts. This may end up not being adequate, but the formula would be more robust, and less of a "sensitive system" as the authors formulate it.

We have implemented this approach to thresholding and have added the results to an additional panel in Figure 27 of the SI. We implemented this by adding two features to the calculation.

For (1) we have counted peaks partially by using a sigmoid function centered on the threshold to assign weights to the peaks. This means that peaks with intensities well below the threshold (<1%) have weights closer to 0, and peaks well above the threshold (100x) are weighted effectively 1.0. Close to the weights assigned to the peaks move slowly between 1.0 (above the threshold) to 0.5 (at the threshold) then down to (0.0) well below the threshold. The mathematical form is: $w(i) = 1.0 / (1 + \exp(-i + t))$, where i is the intensity of the peak and t is the threshold. In place of counting the peaks (n_peaks) we sum the weights for the peaks.

For (2) instead of taking the maximum mass and the minimum mass we take the top 5% and bottom 5% m/z values and average them to replace M_max and M_min respectively. This means that the maximum mass is not just an outlier but instead the end of the distribution of observed masses. Similarly, for the minimum mass.

We found that the numerical value of the mass index has changed but the general trends have not. As an example, we have shown Run 10 here using both calculations.

We have added these results to the SI Section 5.4.

To additional things that the reviewers mentioned in the rebuttal:

- If other referees did not raise the same points as I does not mean that I am wrong. (It does not mean that I am right either). Hence it should not be used as an argument to dismiss my views.

We are pointing out the other experts are happy and they have also seen the comments of this reviewer and have not changed their opinions although we note referee 3 has tried to help bridge the gap.

- I have nothing against a rapid peer-review process. But when, in the first rebuttal, the authors wrote that "urgent publication is needed" and underlined it, it did feel like pressure.

We don't understand this comment. We were merely pointing out that the reviewer was suggesting we should preprint the work before and it felt like they were trying to dismiss one of the key selling points of the work – presentation of a system the community can use to explore their own experiments.

Reviewer #3 (Remarks to the Author):

The authors have already addressed my previous comments in their last response, so I only comment on their response to Reviewer 1. I think R1 and the authors are talking about two different things, and overall, I find the author's response compelling.

We thank the reviewer of referring to our response to reviewer 1 and trying to help bridge the gap.

What R1 says about products being diluted makes sense if it was a linear reaction or several reactions in a linear series. If one assumes each compound can be made by multiple reactions and these reactions do not have to be the same in every run, because in every run the input compounds can change, then it is possible that some compounds will start to dominate if they are a result of several of such processes while other compounds will not.

We agree with reviewer 3 that what reviewer 1 states about dilution could be correct in a linear system and that the design of our system leads to a non-linear reaction mixture. Reviewer 3 has accurately summarized our approach to dominating species in our product mixture and their remarks have been helpful addressing reviewer 1.

The authors are focusing on amplification, which they are defining as persistence of certain products despite cyclic dilutions, while R1 understands amplification as an increasing concentration of a product of a linear $A+B \rightarrow C$ reaction. Each product can be made as a result of several reactions occurring at the same time in a multibranching network. One thing the authors could do to clarify this is to include the discussion from their answers to R1 in the SI, for example (and write explicitly in the main text how they are using the term amplification, although they have somewhat done that already).

Alternatively, they might want to choose a different word than "amplified".

We thank reviewer 3 for clarifying this point. Based on their suggestion we have clarified the statement from the manuscript which now reads: "By recursively cycling and diluting the product mixture with the addition of further starting reagents, our experiments were designed to dilute out any product compounds which are not robust during the reaction cycle. Thus, the only compounds remaining in significant abundance after many cycles would be those that were produced over the cycle, persisted over dilution, or were bound to the mineral surfaces."

We hope we have clarified our results by directly stating what we mean and using persistence over dilution rather than amplification. We have instead added a sentence in the conclusions that suggests amplification could be possible.

We have also added a section about the dilution of certain products to the SI (section 2.1).

We thank reviewer 3 again for helping us approaching these issues and hope that these changes will help to also convince reviewer 1.

Reviewers' Comments:

Reviewer #1:

Remarks to the Author:

In the previous rounds of review I never got the impression that my main comments and concerns had been addressed.

I thank the authors for taking the time and explaining their view in more detail this time.

I do not disregard the fact that the paper has become much better since the first submission.

I appreciate the huge work done by the authors in documenting their research.

From the previous iterations, I acknowledge also all the improvements related to other comments than my three main concerns.

The authors also provided new information addressing some aspects of my main three concerns.

I am happy to now see them addressing the essence of these main concerns in the rebuttal.

Reproducibility:

The reproducibility is still a challenge but I find that it is now addressed better in the manuscript and does not give the impression that the authors claim everything to be reproducible anymore.

Mass index:

This time I find my questions and doubts adequately addressed. I appreciate the additional calculations. I like the figure provided in the rebuttal and the authors could consider adding it to the SI.

Dilution issue:

Let me first comment on the mineral surfaces and detection of adsorbed compounds:

In my understanding, adsorbed compounds will be diluted less strongly because they are not actually in solution. However, if they are not in solution, how can they be detected by the MS (as the MS starts from a liquid sample)? In order for them to be detected, they would need to go in solution, and as a consequence they would be diluted out. Then, wouldn't the adsorbed compounds be nearly invisible in the MS? I struggle to see how a build-up on the mineral surface can lead to a constant/increasing signal for analysis.

This is one of the reasons why I think that the mineral surfaces play little role when discussing the dilution issue in the context of the observed signal, and can be considered to be an edge case. But this is mainly a side note.

Also, I will still consider multiple combinations of reagents producing the same product to be an edge case unless I see some proof that this will happen often with the set of reagents used by the authors. But this is also a side note.

I really like the new figure in the SI 4.2.3. This is what readers like me want to see! This is crucial to prove (or disprove) whether dilution is a problem. How were the four compounds selected?

Now: Does this figure, with the four selected compounds, convince me that dilution is not a problem? No. Here is why:

- Two out of the four compounds, C_3HNO_4 , and $C_3H_5NO_5$, contain a nitrogen atom. They are both present in the cycles 1-10 already. However, in these cycles, the only source of nitrogen would be pyridine, C_5NH_5 . Pyridine being one of the most stable molecules I know, it is likely impossible to break it down to something smaller, i.e. impossible to produce compounds with only 3 carbon atoms. The two compounds, hence, *cannot* have been produced by the added reagents. This indicates that something is going wrong in the values shown in this figure, and as a consequence I cannot believe any other conclusion drawn from this figure.

- A main point of the paper in its current form is the build up of complexity. The abstract even mentions it two times: "the formation of complex molecules", "the production of high complexity molecules". The only molecule out of the four that I would consider to be complex (i.e. needing more than a reaction between two out of the reagents) is $C_{11}H_{10}NO_2$. Sadly, this molecule is diluted out. Hence, I cannot consider this figure as a proof for the build-up of complexity.

For these two reasons, I struggle to give any meaning to the figure as it is.

As of now, it seems to me that the figure in SI 4.2.3 is the main proof advanced by the authors to indicate that the dilution is not an issue; this is mentioned several times in the rebuttal.

In principle, they are right: this is the best way to prove or disprove it.

However, as described above, the current figure shows that something is going wrong, which is why this figure cannot convince me that dilution is not a problem. And hence, without additional prove, I can only stick to my view of a "window of reactivity", and I do not see the proof of the build-up of complexity that is so central to the manuscript as it is.

I am at a loss of what to say because I see the tremendous work accomplished by the authors. But without adequate proof, I cannot agree that there is an actual build-up of complexity that goes beyond a simple window of reactivity. I believe that in the current form, this defeats the main point of the paper.

Reply to reviewer comments. Reviewer comments in italics, our reply in normal type.
REVIEWERS' COMMENTS

Reviewer #1 (Remarks to the Author):

In the previous rounds of review I never got the impression that my main comments and concerns had been addressed.

We are sorry for this; we did address everything point by point and attempt to clarify things according to the comments but we think there might be a conceptual mismatch.

I thank the authors for taking the time and explaining their view in more detail this time.

We are glad that the reviewer thinks we have explained things to them better.

I do not disregard the fact that the paper has become much better since the first submission.

I appreciate the huge work done by the authors in documenting their research.

We agree, and hope that the design of the platform, and the concept of exploring chemical space in this way is really important to make progress. Also I think that given the reviewer was happy with our platform is also very good.

From the previous iterations, I acknowledge also all the improvements related to other comments than my three main concerns. The authors also provided new information addressing some aspects of my main three concerns. I am happy to now see them addressing the essence of these main concerns in the rebuttal.

We are glad the reviewer appreciates our efforts taken and sees the improvements in the manuscript and their concerns addressed.

Reproducibility:

The reproducibility is still a challenge but I find that it is now addressed better in the manuscript and does not give the impression that the authors claim everything to be reproducible anymore.

We never said everything was reproducible except the main trends. The key thing is that complex chemical systems are very complicated and just as non-linear dynamic systems, there is a sensitive dependence on initial conditions. That is why it is important to look for general trends and attractors. This was been consistent the entire time.

Mass index:

This time I find my questions and doubts adequately addressed. I appreciate the additional calculations. I like the figure provided in the rebuttal and the authors could consider adding it to the SI.

We are glad that the doubts of reviewer are now addressed and the figure has been added to the SI section 5.4.

Dilution issue:

Let me first comment on the mineral surfaces and detection of adsorbed compounds:

In my understanding, adsorbed compounds will be diluted less strongly because they are not actually in solution. However, if they are not in solution, how can they be detected by the MS (as the MS starts from a liquid sample)? In order for them to be detected, they would need to go in solution, and as a consequence they would be diluted out. Then, wouldn't the adsorbed compounds be nearly invisible in the MS? I struggle to see how a build-up on the mineral surface can lead to a constant/increasing signal for analysis.

Whilst we understand the struggles of the reviewer, the data says something different and I think this will be a very important set of studies to build on.

This is one of the reasons why I think that the mineral surfaces play little role when discussing the dilution issue in the context of the observed signal, and can be considered to be an edge case. But this is mainly a side note.

We agree with the reviewer that a compound adsorbed on to the mineral surface would not be detected by the MS but a change of the input composition, and therefore of the reaction environment, could lead to this compound going back into solution and being detected later in the same experiment. This effect could then have quite a big role in the overall development of the experiment.

Also, I will still consider multiple combinations of reagents producing the same product to be an edge case unless I see some proof that this will happen often with the set of reagents used by the authors. But this is also a side note.

The network model has shown, as described in the manuscript, that 2206 reactions have been predicted. The number of products from those reactions is 429. This is proof that multiple reactions can produce the same product and it is therefore not an edge case.

I really like the new figure in the SI 4.2.3. This is what readers like me want to see! This is crucial to prove (or disprove) whether dilution is a problem. How were the four compounds selected?

We are glad that the reviewer likes the figure. As described in the previous reply, the compounds were taken from table 5 in the SI section 4.2.2. These formulas have been found by comparing the product formula from the reaction network with the proposed formula for the experimentally found m/z values shown in SI section 4.2.1.

Now: Does this figure, with the four selected compounds, convince me that dilution is not a problem? No. Here is why:

*- Two out of the four compounds, C₃HNO₄, and C₃H₅NO₅, contain a nitrogen atom. They are both present in the cycles 1-10 already. However, in these cycles, the only source of nitrogen would be pyridine, C₅NH₅. Pyridine being one of the most stable molecules I know, it is likely impossible to break it down to something smaller, i.e. impossible to produce compounds with only 3 carbon atoms. The two compounds, hence, *cannot* have been produced by the added reagents. This indicates that something is going wrong in the values shown in this figure, and as a consequence I cannot believe any other conclusion drawn from this figure.*

These formulae are tentative assignments from experimentally observed m/z values. As described in the SI section 4.2.1, these formulae are only tentatively proposed, and as they are experimentally found from a messy chemistry mixture, should only be taken as suggestions as to what might be in the mixture. This also means that these compounds could be very likely very different, this is part of the challenge of analysing complex chemical mixtures with MS, the challenge discussed in the manuscript. While the formula was added to the figure for illustration, it was not the data used to track these peaks. The product peaks have been tracked by their m/z values, which are clear values matching and possible to track, therefore the figure is correct. We have removed the proposed formulas from the figure for clarity and updated it in the SI.

- A main point of the paper in its current form is the build up of complexity. The abstract even mentions it two times: "the formation of complex molecules", "the production of high complexity molecules". The only molecule out of the four that I would consider to be complex (i.e. needing more than a reaction between two out of the reagents) is C₁₁H₁₀NO₂. Sadly, this molecule is diluted out. Hence, I cannot consider this figure as a proof for the build-up of complexity.

For these two reasons, I struggle to give any meaning to the figure as it is.

We disagree with the reviewer in this point, even when the concentration of C₁₁H₁₀NO₂ is decreasing over time, it is a complex molecule (as the reviewer already stated above) made by this system. It was not added by us and was not a contaminant, so observing it as we have proves that this unconstrained system with random input compositions can produce complex molecules.

As of now, it seems to me that the figure in SI 4.2.3 is the main proof advanced by the authors to indicate that the dilution is not an issue; this is mentioned several times in the rebuttal.

In principle, they are right: this is the best way to prove or disprove it. However, as described above, the current figure shows that something is going wrong, which is why this figure cannot convince me that dilution is not a problem. And hence, without additional prove, I can only stick to my view of a "window of reactivity", and I do not see the proof of the build-up of complexity that is so central to the manuscript as it is.

I am at a loss of what to say because I see the tremendous work accomplished by the authors. But without adequate proof, I cannot agree that there is an actual build-up of complexity that goes beyond a simple window of reactivity. I believe that in the current form, this defeats the main point of the paper.

Persistence through cycles and the production of complex molecules has been demonstrated as shown above. These reviews have been tremendously useful and have allowed us to improve our manuscript a great deal, and we have shown the reviewer direct proof, but the reviewer seems to be a little blind to it. This cognitive dissidence is common when new results challenge old ideas, and we think the reviewer has been as fair and open as they can. Also, we deeply appreciate the amount of time they have spent on their comments. Whilst we don't entirely agree, we respect this considerable effort and have tried to answer these comments openly and constructively.